# Deep learning-based cross-classifications reveal conserved spatial behaviors within tumor histological images

Javad Noorbakhsh[1,7], Saman Farahmand [2,7], Ali Foroughi pour[1,7], Sandeep Namburi [1], Dennis Caruana[3], David Rimm [3], Mohammad Soltanieh-ha[4], Kourosh Zarringhalam[2,5] & Jeffrey H. Chuang [1,6 ✉]

Histopathological images are a rich but incompletely explored data type for studying cancer. Manual inspection is time consuming, making it challenging to use for image data mining. Here we show that convolutional neural networks (CNNs) can be systematically applied across cancer types, enabling comparisons to reveal shared spatial behaviors. We develop CNN architectures to analyze 27,815 hematoxylin and eosin scanned images from The Cancer Genome Atlas for tumor/normal, cancer subtype, and mutation classification. Our CNNs are able to classify TCGA pathologist-annotated tumor/normal status of whole slide images (WSIs) in 19 cancer types with consistently high AUCs (0.995 ± 0.008), as well as subtypes with lower but significant accuracy (AUC 0.87 ± 0.1). Remarkably, tumor/normal CNNs trained on one tissue are effective in others (AUC 0.88 ± 0.11), with classifier relationships also recapitulating known adenocarcinoma, carcinoma, and developmental biology. Moreover, classifier comparisons reveal intra-slide spatial similarities, with an average tile-level correlation of 0.45 ± 0.16 between classifier pairs. Breast cancers, bladder cancers, and uterine cancers have spatial patterns that are particularly easy to detect, suggesting these cancers can be canonical types for image analysis. Patterns for TP53 mutations can also be detected, with WSI self- and cross-tissue AUCs ranging from 0.65-0.80. Finally, we comparatively evaluate CNNs on 170 breast and colon cancer images with pathologist-annotated nuclei, finding that both cellular and intercellular regions contribute to CNN accuracy. These results demonstrate the power of CNNs not only for histopathological classification, but also for cross-comparisons to reveal conserved spatial behaviors across tumors.

[1] The Jackson Laboratory for Genomic Medicine, Farmington, CT, USA. [2] Computational Sciences PhD Program, University of Massachusetts-Boston, Boston, MA, USA. [3] Department of Pathology, Yale University School of Medicine, New Haven, CT, USA. [4] Department of Information Systems, Boston University, Boston, MA, USA. [5] Department of Mathematics, University of Massachusetts-Boston, Boston, MA, USA. [6] UCONN Health, Department of Genetics and Genome Sciences, Farmington, CT, USA. [7] These authors contributed equally: Javad Noorbakhsh, Saman Farahmand, Ali Foroughi pour. ✉email: Jeff.Chuang@jax.org

Histopathological images are a crucial data type for diagnosis of cancer malignancy and selecting treatment[1], indicative of their value for understanding cancer biology. However, manual analysis of whole-slide images (WSIs) is labor-intensive[2] and can vary by observer[3–5], making it difficult to scale such approaches for discovery-oriented analysis of large image collections. Image datasets for hematoxylin and eosin (H&E), immunohistochemistry (IHC), and spatial -omic imaging technologies are rapidly growing[6]. Improved computational approaches for analyzing cancer images would therefore be valuable, not only for traditional tasks such as histopathological classification and cell segmentation[7] but also for novel questions such as the de novo discovery of spatial patterns that distinguish cancer types. The search for recurrent spatial patterns is analogous to the search for common driver mutations or expression signatures based on cancer sequencing[8], yet this paradigm has been little explored for cancer image data.

In the last few years, there have been major advances in supervised and unsupervised learning in computational image analysis and classification[6,9], providing opportunities for application to tumor histopathology. Manual analysis involves assessments of features such as cellular morphology, nuclear structure, or tissue architecture, and such pre-specified image features have been inputted into support vector machines or random forests for tumor subtype classification and survival outcome analysis e.g.,[10–12]. However, pre-specified features may not generalize well across tumor types, so recent studies have focused on fully-automated approaches using convolutional neural networks (CNNs), bypassing the feature specification step. For example, Schaumberg et al. trained ResNet-50 CNNs to predict SPOP mutations using WSIs from 177 prostate cancer patients[13], achieving AUC = 0.74 in cross-validation and AUC = 0.64 on an independent set. Yu et al. utilized CNN architectures including AlexNet, GoogLeNet, VGGNet-16[14], and the ResNet-50 to identify transcriptomic subtypes of lung adenocarcinoma (LUAD) and squamous cell carcinoma (LUSC)[15]. They were able to classify LUAD vs. LUSC (AUC of 0.88–0.93), as well as each vs. adjacent benign tissues with higher accuracy. Moreover, they were able to predict the TCGA transcriptomic classical, basal, secretory, and primitive subtypes of LUAD[16,17] with AUCs 0.77–0.89, and similar subtype classifications have been reported in breast[18]. Recently, Coudray et al.[19] proposed a CNN based on Inception v3 architecture to classify WSIs in LUAD and LUSC, achieving an AUC of 0.99 in tumor/normal classification. Further, their models were able to predict mutations in ten genes in LUAD with AUCs 0.64–0.86, and subsequently mutations in BRAF (AUC ~0.75) or NRAS (AUC ~0.77) melanomas[20]. Other groups have used CNNs to distinguish tumors with high or low mutation burden[21]. These advances highlight the potential of CNNs in computer-assisted analysis of WSIs.

Many critical questions remain. For example, prior studies have focused on individual cancer types, but there has been little investigation of how neural networks trained on one cancer type perform on other cancer types, which could provide important biological insights. As an analogy, comparisons of sequences from different cancers have revealed common driver mutations[22,23], e.g., both breast and gastric cancers have frequent HER2 amplification, and both are susceptible to treatment by trastuzumab[24,25]. Such analysis is in a rudimentary state for image data, as it remains unclear how commonly spatial behaviors are shared between cancer types. A second important question is the impact of transfer learning on cancer image analysis, which has been used in studies such as Coudray et al.[19]. Transfer learning is used to pre-train neural networks using existing image compilations Zhu et al.[26]. However, standard compilations are not histological, and it is unclear how this affects

cancer studies. A third key topic is to clarify the features that impact prediction accuracy. For example, recurrent neural network approaches[27] have been shown to distinguish prostate, skin, and breast cancers at the slide level, but the relevant spatial features are not well-understood. Determination of predictive features is affected not only by the underlying biology but also by the availability of spatial annotations and appropriate computational techniques.

To investigate these questions, here we analyze 27,815 frozen or FFPE whole-slide H&E images from 23 cancer types from The Cancer Genome Atlas (TCGA), a resource with centralized rules for image collection, sequencing, and sample processing. We have developed image processing and convolutional neural network software that can be broadly applied across tumor types to enable cross-tissue analyses. Using these techniques, first, we show that this CNN architecture can distinguish tumor/normal and cancer subtypes in a wide range of tissue types. Second, we systematically compare the ability of neural networks trained on one cancer type to classify images from another cancer type. We show that cross-classification relationships recapitulate known tissue biology. Remarkably, these comparisons also reveal that breast, bladder, and uterine cancers can be considered canonical cancer image types. Third, we investigate driver effects by determining how cancers with the TP53 mutation can be cross-classified across tissues, including a comparison of transfer learning vs. full CNN training. Fourth, we test how cellular vs. intercellular regions impact CNN tumor/normal predictions, making use of cell-resolution annotations from 170 colorectal and breast cancer images. Our studies demonstrate that cross-comparison of CNN classifiers is a powerful approach for discovering shared biology within cancer images.

## Results

**Pan-cancer convolutional neural networks for tumor/normal classification**. We developed a CNN architecture to classify slides from TCGA by tumor/normal status, using a neural network that feeds the last fully connected layer of an Inception v3-based CNN pretrained on ImageNet into a fully connected layer with 1024 neurons. This architecture is depicted in Fig. 1a, and a related architecture for mutation classification (described in sections below) is shown for comparison in Fig. 1b. The two final fully connected layers of the tumor/normal CNN were trained on tiles of size $512 \times 512$ from WSIs. Due to insufficient FFPE normal WSIs in TCGA, for this task, we only used flash-frozen samples. We trained this model separately on slides from 19 TCGA cancer types having numbers of slides ranging from 205 to 1949 (Fig. 2a). In all, 70% of the slides were randomly assigned to the training set and the rest were assigned to the test set. To address the data imbalance problem[28], the majority class was undersampled to match the minority class.

Figure 2b shows the classification results. We used a naive training approach such that, after removal of background regions, all tiles in a normal image are assumed normal and all tiles in a tumor image are assumed tumor. The CNN accurately classifies test tiles for most tumor types (accuracy: $0.91 \pm 0.05$, precision: $0.97 \pm 0.02$, recall: $0.90 \pm 0.06$, specificity: $0.86 \pm 0.07$. Mean and standard deviation calculated across cancer types). We next examined the fraction of tiles classified as tumor or normal within each slide. The fractions of tiles matching the slide annotation are $0.88 \pm 0.14$ and $0.90 \pm 0.13$ for normal and tumor samples, respectively (Fig. 2c) (mean and standard deviation calculated from all cancer types pooled together). These fractions are high in almost all slides, and the tumor-predicted fraction (TPF) is significantly different between tumors and normals ($P < 0.0001$ per-cohort comparison of tumor vs. normal, Welch's $t$ test). We

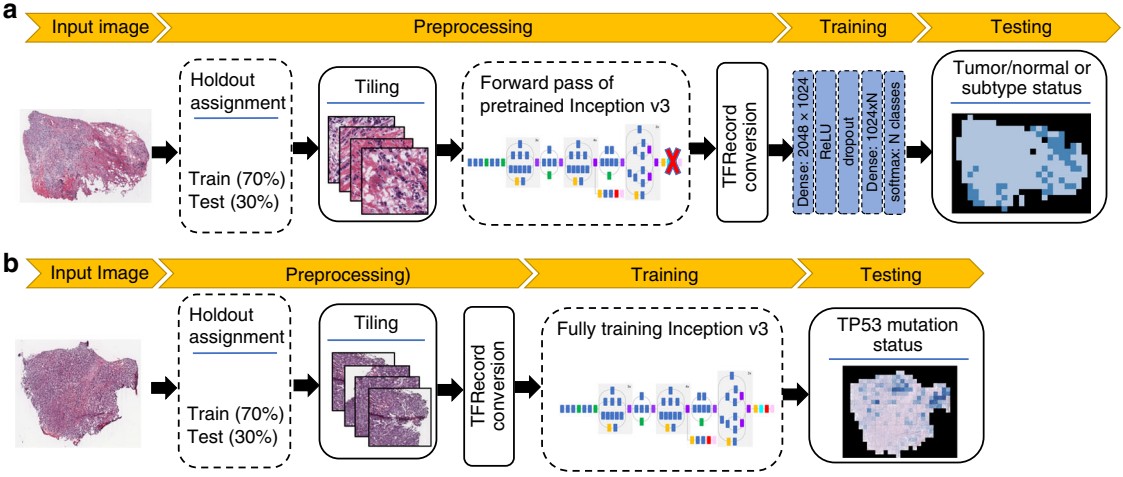

**Fig. 1 Classification pipelines. a** Transfer-learning pipeline for tumor/normal and subtype classification. **b** Full training pipeline for mutation classification.

**Fig. 2 Tumor/normal classification using CNNs. a** Numbers of tumor and normal slides in test and training sets. **b** Per-tile classification metrics. Tile-level test set sizes are provided in Supplementary Data 3. **c** Fraction of tiles predicted to be tumor within each slide. **d** Per-slide AUC values for tumor/normal classification for ROC and precision-recall curve (PR). **e** Pearson correlation coefficients between predicted and pathologist evaluation of tumor purity. *P* values are based on the permutation test of the dependent variable after Bonferroni correction across all cancer types. Raw and adjusted *P* values are provided in Supplementary Data 4.

also performed the classification on a per-slide basis. To do this, we used the TPF in each slide as a metric to classify it as tumor or normal. This approach yielded extremely accurate classification results for all cancer types (Fig. 2d, mean AUC ROC = 0.995, mean PR AUC = 0.998). Confidence intervals (CI) of per-slide predictions are given in Supplementary Fig. 1a (also see "Methods"). The CI lower bound on all classification models was above 90%, with cancer types having fewer slides or imbalanced test data tending to have larger CIs. These results indicate that our network can successfully classify WSIs as tumor/normal across many different cancer types. These results were for slide-level test/train splits of the data, but splitting at the patient level instead had little effect on classification accuracy (see "Methods" and Supplementary Fig. 1b). Most misclassification was for adjacent normal slides with unexpectedly large predictions for TPF. Manual pathology review indicated that such slides often suffer from poor quality, tissue folding, or excessive tissue damage related to freezing (e.g., see Supplementary Fig. 3).

We next investigated if TPF correlates with tumor purity, that is, slides with higher tumor purity tend to have larger TPFs and vice versa. We found significant positive correlations between TPF and TCGA pathologist-reported purity ("average percentage of tumor cells") in the majority of cancer types (Fig. 2e), with larger cancer sets tend to have more significant p values (e.g., BRCA: $P = 5e\text{-}17$). The distributions of TPF were systematically higher than the pathologist annotations (Supplementary Fig. 2), though this difference can be partially reconciled by the fact that TPF is based on the neoplastic area while the pathologist annotation is based on cell counts. Tumor cells are larger than stromal cells and reduce nuclear density. While TPF and purity are clearly related, the moderate magnitudes of correlations indicate that intraslide improvements can be made. A notable limitation is the training assumption that tiles in a slide are either all tumor or all normal, as intraslide pathologist annotations are not provided by TCGA. Additionally, pathologist assessments of tumor purity have non-negligible variability[29] that may affect correlations. For comparison, we also calculated the correlation of TCGA pathologist-reported purity with the genomics-inferred purity measures ABSOLUTE[30] and InfiniumPurify[31] in BRCA. The correlations of TCGA-annotated purity vs. the ABSOLUTE and InfiniumPurify estimates were only 0.16 and 0.10, respectively. These correlations were lower than our observed correlation between TPF and purity ($r \sim 0.4$).

**Neural network classification of cancer subtypes**. We also applied our algorithm to classify tumor slides based on their cancer subtypes (Fig. 1a). This analysis was performed on ten tissues for which pathologist subtype annotation was available on TCGA: sarcoma (SARC), brain (LGG), breast (BRCA), cervix (CESC), esophagus (ESCA), kidney (KIRC/KIRP/KICH), lung (LUAD/LUSC), stomach (STAD), uterine (UCS/UCEC), and testis (TGCT). Cancer subtypes with at least 15 samples were considered, based on TCGA metadata (see "Methods"). Because comparable numbers of FFPE and flash-frozen samples are present in TCGA cancer types (FFPE to frozen slide ratio: 1.0 ± 0.5), both were included (Fig. 3a), and each tissue was stratified into its available subtypes (Fig. 3b and "Methods"). We used the same CNN model as for tumor/normal classification; however, for cancer types with more than two subtypes, a multi-class classification was used.

Figure 3c, d shows the per-tile and per-slide classification results (AUC ROCs alongside their micro- and macro-averages). At the slide level, the classifiers can identify the subtypes with good accuracy in most tissues, though generally not yet at clinical precision (AUC micro-average: 0.87 ± 0.1; macro-average: 0.87 ±

0.09). The tissue with the highest AUC micro/macro-average was kidney (AUC 0.98), while the lowest was a brain with micro-average 0.60 and macro-average 0.67. All CIs were above the 0.50 null AUC expectation, and all of the AUCs were statistically significant (5% FDR, Benjamini–Hochberg correction[32]). For full CIs and $P$ values, see Supplementary Data 1. The individual subtype with the best AUC is the mucinous subtype for breast cancer (adjusted $P$ value <1e-300). The weakest $P$ value (adjusted $P = 0.012$) belongs to the oligoastrocytoma subtype of the brain. Slide predictions are superior to those at the tile level, though with similar trends across tissues. This indicates that tile averaging provides substantial improvement of signal to noise, consistent with observations for the tumor/normal analysis. In contrast to tumor/normal classification achieving high AUC's across all cancer types, subtype classification AUCs are lower and span a wider range. This suggests that subtype classification is inherently more challenging than tumor/normal classification, with a narrower range of image phenotypes.

The images used in the subtype analysis were from a mixture of frozen and FFPE samples. Although FFPE samples are preferred because they avoid distortions caused by freezing, we tested whether the CNNs were able to classify subtypes for each sample preparation (Supplementary Fig. 4). The CNNs classified the FFPE and frozen samples with comparable accuracy, with the same tumor types doing better (e.g., kidney), or worse (brain) in each. Correlations between classification AUCs were high across the two sample preparations ($r = 0.87$ for macro-averages; $r = 0.78$ for micro-averages). As expected, FFPE-based classifications were generally better, notably for brain and sarcoma samples.

**Cross-classifications between tumor types demonstrate conserved spatial behaviors**. We next used cross-classification to test the hypothesis that different tumor types share CNN-detectable morphological features distinct from those in normal tissues. For each cancer type, we re-trained the binary CNN classifier for tumor/normal status using all flash-frozen WSIs in the set. We then tested the ability of each classifier to predict tumor/normal status in the samples from each other cancer type. Figure 4 shows a heatmap of per-slide AUC for all cross-classifications, hierarchically clustered on the rows and columns of the matrix. A non-clustered version is presented in Supplementary Fig. 5 with CIs. Surprisingly, neural networks trained on any single tissue were successful in classifying cancer vs. normal in most other tissues (average pairwise AUCs of off-diagonal elements: 0.88 ± 0.11 across all 342 cross-classifications). This prevalence of strong cross-classification supports the existence of morphological features shared across cancer types but not normal tissues. In particular, classifiers trained on most cancer types successfully predicted tumor/normal status in BLCA (AUC = 0.98 ± 0.02), UCEC (AUC = 0.97 ± 0.03), and BRCA (AUC = 0.97 ± 0.04), suggesting that these cancers most clearly display features universal across types. At a 5% FDR, 330 cross-classification AUCs are significant (See Supplementary Fig. 5 for statistical details). The AUC mean and CI lower bound are each above 80% for 300 and 164 of these cross-classifications, respectively. A few cancer types, e.g., LIHC and PAAD, showed poor cross-classification to other tumor types, suggesting morphology distinct from other cancers.

To improve spatial understanding of these relationships, we tested how well tile-level predictions are conserved between different classifiers (Fig. 5), while also analyzing the effect of varying the test set. For each pair of classifiers, we specified a test set then computed the correlation coefficient of the predicted tumor/normal state (logit of the tumor probability) across all tiles in the test set. We repeated this calculation for each test set, which

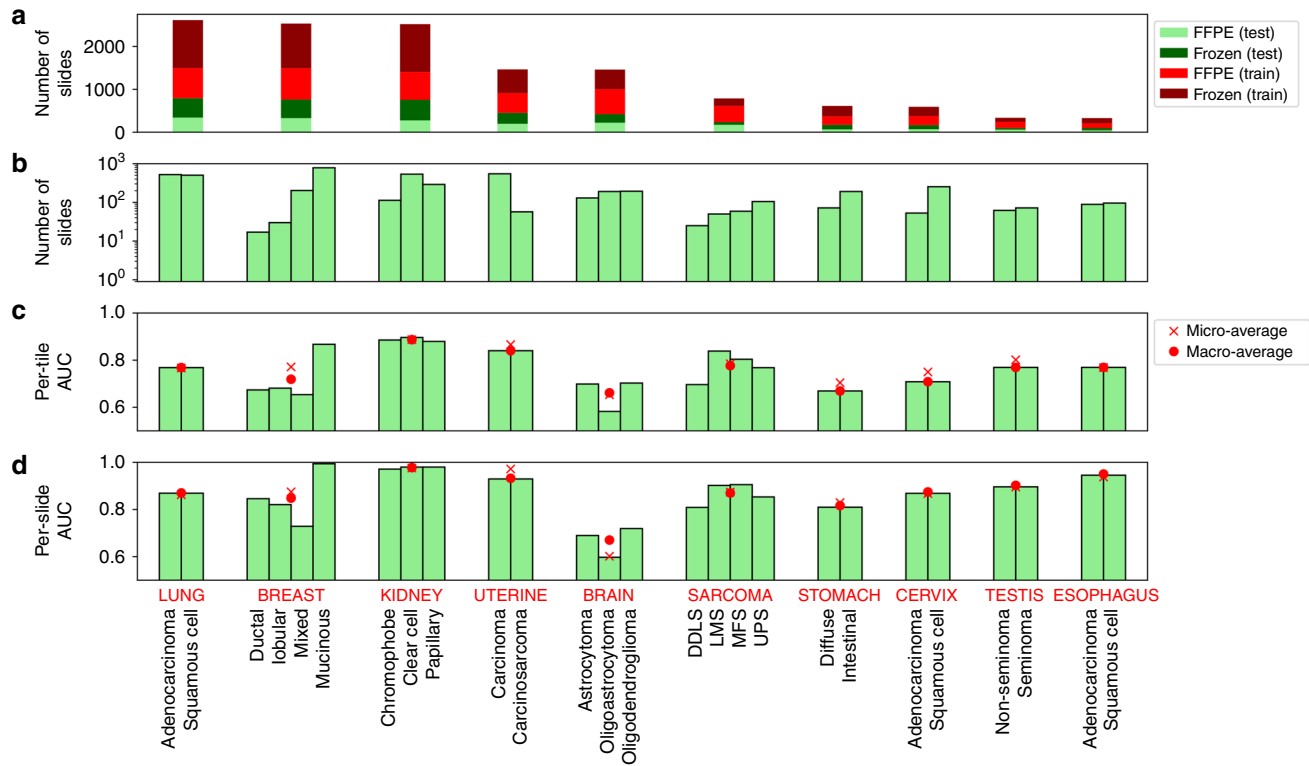

**Fig. 3 Subtype classification using CNNs. a** The number of samples used for training. **b** The number of samples for each subtype. **c** AUC ROC for subtype classifications at the tile level (**d**) and at the slide level.

we indexed by tissue type (breast, bladder, etc.). Each test set included both tumor and normal slides for the tissue type. Figure 5a, b shows for each pair of classifiers the average and maximum correlation coefficients, respectively, over test sets. Many correlations are positive, with an average and standard deviation over all pairs of classifiers of 45 ± 16% (Fig. 5a, diagonal elements excluded), indicating cross-classifiers agree at the tile level. These tile-level results supported the slide-level results. Classifiers with low cross-classification slide-level AUCs, such as LIHC, had the smallest tile-level correlations. Tile predictions also showed similarities between classifiers derived from the same tissue (e.g., LUAD-LUSC, KICH-KIRP-KIRC). Similarities between classifiers became even more apparent when we focused on the test tissue with the strongest correlation for each pair of classifiers (Fig. 5b). These positive correlations are not simply due to distinguishing tiles in tumor slides from tiles in normal slides. Figure 5c, d is analogous to Fig. 5a, b, but computed only over tumor slides. The results are nearly unchanged, indicating that they reflect behavior within tumor images.

We hypothesized that certain tissue types might be particularly easy to classify, and to test this we tabulated which tissue sets yielded the maximal correlations for each pair of classifiers in Fig. 5b (Supplementary Data 2). For each pair, we listed the three tissue sets yielding the highest correlations. If this were random, we would expect each tissue to appear in this list 27 times. However, we observed extreme prevalence for BRCA (132 appearances, $P = 8.5e\text{-}119$), BLCA (106 appearances, $P = 2.5e\text{-}43$), and UCEC (62 appearances, $P = 1.9e\text{-}11$). Many classifier pairs agree better within these three tissues than they do within their training tissues. Thus BRCA, BLCA, and UCEC are canonical types for intraslide spatial analysis, in addition to their high cross-classifiability at the whole-slide level (Fig. 4).

We compared the effect of minor modification to the architecture on tumor/normal self- and cross-classifications. If we just used the

Inception v3 architecture without the additional dense layers (see "Methods"), the results were inferior (Supplementary Fig. 6). Our architecture (Fig. 1a) achieved a slightly higher AUC on average (0.04 ± 0.068) compared to the original Inception V3 network (Wilcoxon signed-rank test $P$ value <1e-20).

**Cross-classification relationships recapitulate cancer tissue biology.** To test the biological significance of cross-classification relationships, we assessed associations between the tissue of origin[22] and cross-classification clusters. Specifically, we labeled KIRC/KIRP/KICH as pan-kidney[33], UCEC/BRCA/OV as pan-gynecological (pan-gyn)[34], COAD/READ/STAD as pan-gastrointestinal (pan-GI)[35], and LUAD/LUSC as lung. The hierarchical clustering in Fig. 4 shows that cancers of similar tissue of origin cluster closer together. We observed that the lung set clusters together on both axes, Pan-GI clusters on the test and partially the train axis, and Pan-Gyn also partially clusters on the test axis. Pan-Kidney partially clusters on both axes. To quantify this, we tested the associations between proximity of cancers on each axis and similarity of their phenotype (i.e., tissue of origin/ adeno-ness). Organ of origin was significantly associated with smaller distances in the hierarchical clustering ($P$ value = 0.002 for test axis and $P = 0.009$ for train axis; Gamma index permutation test, see "Methods"). We also grouped cancers by adeno-carcinoma/carcinoma status (Fig. 4, second row from top). Since SARC does not fit either category, and ESCA contains a mixture of both categories, these two cancers were labeled as "other". The inter-cancer distances were significantly associated with adeno-ness on the train axis ($P$ value = 0.015). We observed other intriguing relationships among cross-tissue classifications as well. Particularly, Pan-GI created a cluster with Pan-Gyn, supporting these tumor types having shared features related to malignancy. Likewise, Pan-Kidney and lung also cluster close to each other.

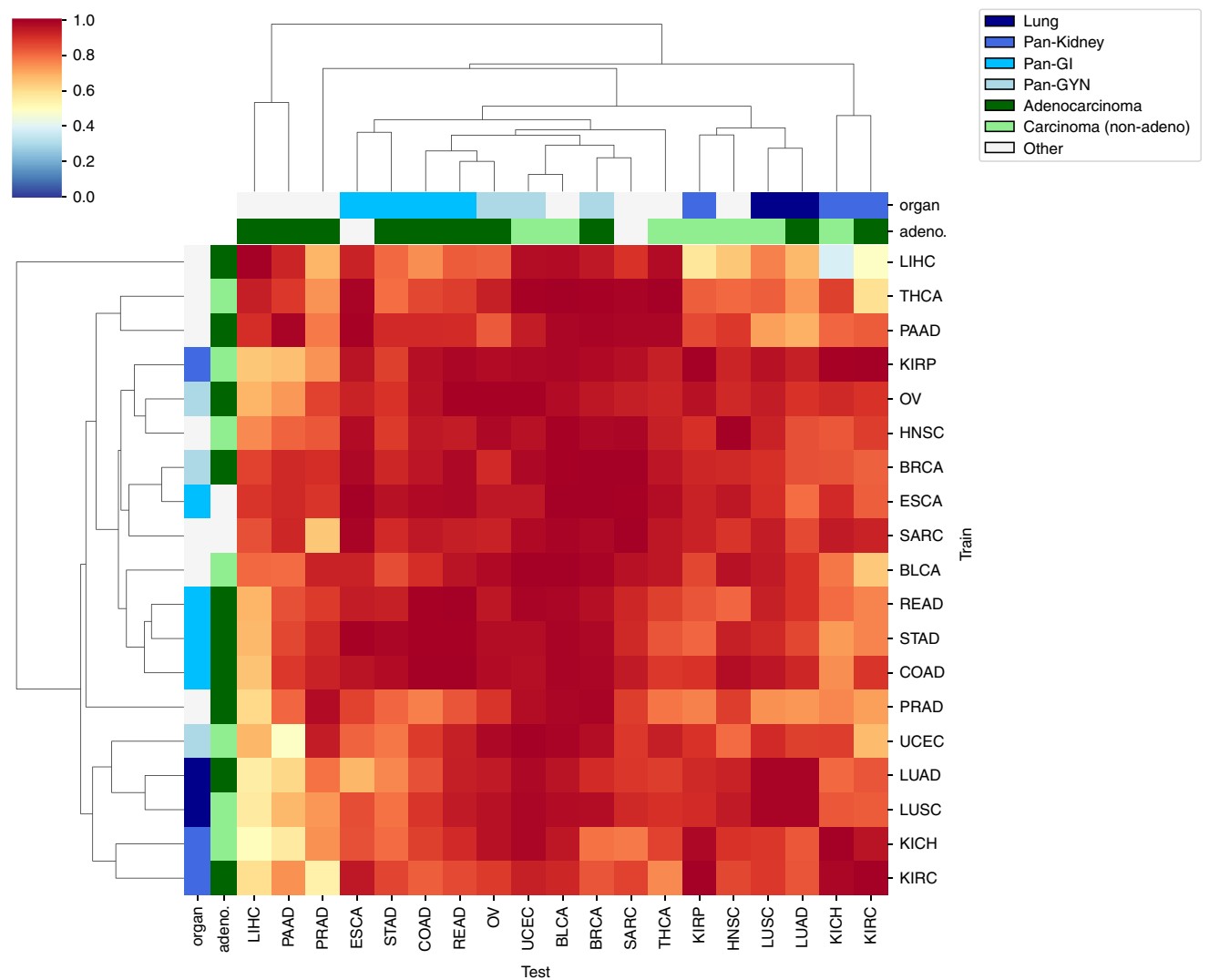

**Fig. 4 Per-slide AUC values for cross-classification of tumor/normal status.** The hierarchically clustered heatmap shows pairwise AUC values of CNNs trained on the tumor/normal status of one cancer type (train axis) tested on the tumor/normal status of another cancer type (test axis). Adeno-ness (adenocarcinoma vs. non-adenocarcinoma) and organ of origin (lung, kidney, gastrointestinal, gynecological) for each set are marked with colors on the margins. Cancers with ambiguous or mixed phenotype are marked as "Other".

**Validation of cross-classification relationships using CPTAC images.** To validate the trained CNNs and their cross-classification accuracies, we applied them to the LUAD and LUSC slides of the Clinical Proteomic Tumor Analysis Consortium (CPTAC) dataset (see "Methods"). TCGA-trained LUAD and LUSC classifiers were highly effective on the CPTAC LUAD and LUSC datasets (Fig. 6a, b). The TCGA-trained LUAD and LUSC classifiers have validation AUCs of 0.97 and 0.95, respectively, on the CPTAC-LUAD dataset, and have validation AUCs of 0.97 and 0.98, respectively, on the CPTAC-LUSC dataset. Both of the TCGA-trained CNNs yielded well-separated distributions of TPF between CPTAC tumor and normal slides (Supplementary Fig. 7). CNNs trained on other TCGA tissue types were also relatively effective on the CPTAC sets, with average AUC of 0.75 and 0.73 when applied to the CPTAC-LUAD and LUSC image sets, respectively. This was lower than the performance of the TCGA-trained classifiers on the TCGA LUAD and LUSC sets (average AUC 0.85 and 0.90, respectively), suggesting that cross-classification is more sensitive to batch protocol variations. However, the correlation between AUCs on the TCGA and CPTAC sets was high (Fig. 6c, d: LUAD: $r = 0.90$, LUSC: $r = 0.83$), indicating that relationships

between tumor types have a clear signal despite such sensitivities. The CPTAC-LUAD and LUSC datasets were also used to train classifiers, which were then tested on the TCGA cancer sets. We observed high correlation between TCGA-trained and CPTAC-trained cross-classification AUCs (Supplementary Fig. 8, LUAD: $r = 0.98$, LUSC: $r = 0.90$).

**Comparisons of neural networks for TP53 mutation classification.** To investigate how images can be used to distinguish cancer drivers, we tested the accuracy of CNNs for classifying TP53 mutation status in five TCGA cancer types, namely BRCA, LUAD, STAD, COAD, and BLCA. We chose these due to their high TP53 mutation frequency[36–38], providing sufficient testing and training sets for cross-classification analysis. Using transfer learning, we obtained moderate to low AUCs for TP53mut/wt classification (0.66 for BRCA, 0.64 for LUAD, 0.56 for STAD, 0.56 for COAD, and 0.61 for BLCA). Due to this weak performance, we switched to a more computationally intensive approach in which we fully trained all parameters of the neural networks based on an architecture described in ref. [19] (Fig. 1b), with undersampling to address data imbalance and a 70/30 ratio

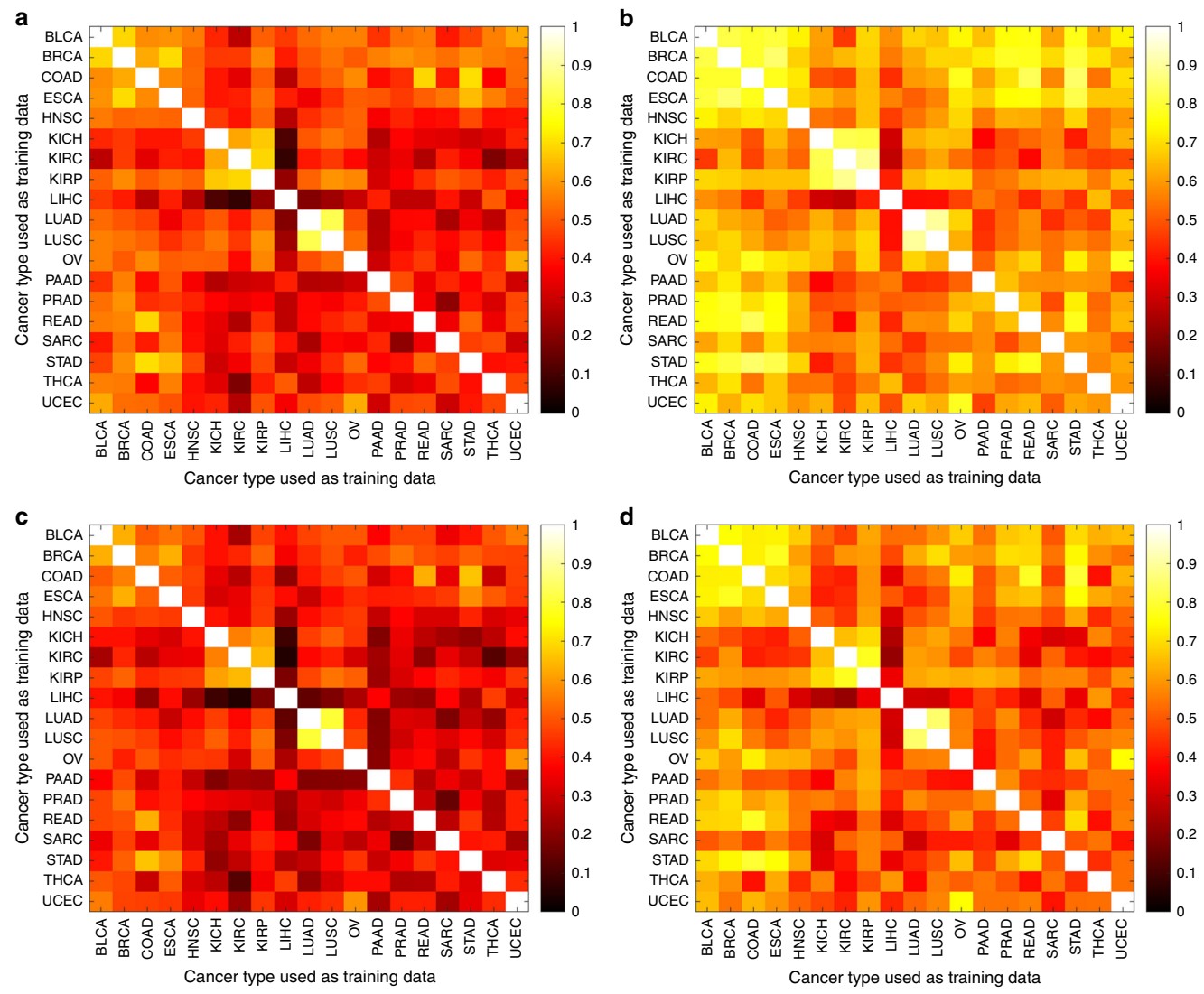

**Fig. 5 Tile-level cross-classifications as a function of test set.** Correlations of predicted tumor/normal status (i.e., logit of tumor probability) between pairs of classifiers, specified on the x and y axis. Correlations are first calculated using the tile values for all slides of a given test tissue. **a** Average correlation across tissues, using both tumor and normal slides in the tissue test sets. **b** Correlation for the tissue set with the maximal correlation, using both tumor and normal slides in the tissue test sets. **c** Average correlation across tissues, using only tumor slides in the tissue test sets. **d** Correlation for the tissue set with the maximal correlation, using only tumor slides in the tissue test sets.

of slides for training and testing. Figure 7a, b shows heatmaps of AUC for the per-tile and per-slide classification results, respectively (see also Supplementary Figs. 9 and 10). Self-cohort predictions (diagonal values) have AUC values ranging from 0.65–0.80 for per-slide and 0.63–0.78 for per-tile evaluations. Stomach adenocarcinoma (slide AUC = 0.65) was notably more difficult to predict than lung adenocarcinoma (slide AUC = 0.80), for which we found AUC values comparable to the AUC = 0.76 LUAD results reported by Coudray et al.[19]. This LUAD fully trained network (AUC = 0.76) outperformed the transfer learning for the same data (AUC = 0.64). The CNNs achieved a higher AUC compared with a random forest using tumor purity and stage for TP53 mutation prediction (see Supplementary Fig. 11), suggesting the CNNs use more sophisticated morphological features in their predictions. We also observed that CNNs were able to more accurately identify tumors with TP53 mutations when the allele frequency of the mutation was higher, suggesting that prediction is easier when the tumor is more homogeneous (Supplementary Fig. 12). The F1 scores of the CNNs are provided in Supplementary Fig. 13.

We also tested the ability of the TP53 CNNs to cross-predict across cancer types. Cross-predictions yielded AUC values with a comparable range as the self-cohort analyses (AUCs 0.62–0.72 for slides; 0.60–0.70 for tiles), though self-cohort analyses were slightly more accurate. These AUC values are not sufficient for practical use, though the positive cross-classification results suggest that it might be possible to combine datasets to increase accuracy (see "Discussion"). Colon adenocarcinoma AUC values tended to be low as both a test and train set, suggesting TP53 creates a different morphology in this tissue type. Overall, the positive cross-classifiabilities support the existence of shared TP53 morphological features across tissues. Figure 8 shows TP53 mutational heatmaps of one LUAD slide known to be mutant and one LUAD slide known to be wild type from the sequencing data. We compared the LUAD- and BRCA-trained deep learning models on these slides, as those two models provided the highest AUC values in our cross-classification experiments. Prediction maps for tumor/normal status (second row) and TP53 mutational status (third row) are shown for both samples. Both tumor/normal models correctly predicted the majority of tiles in each

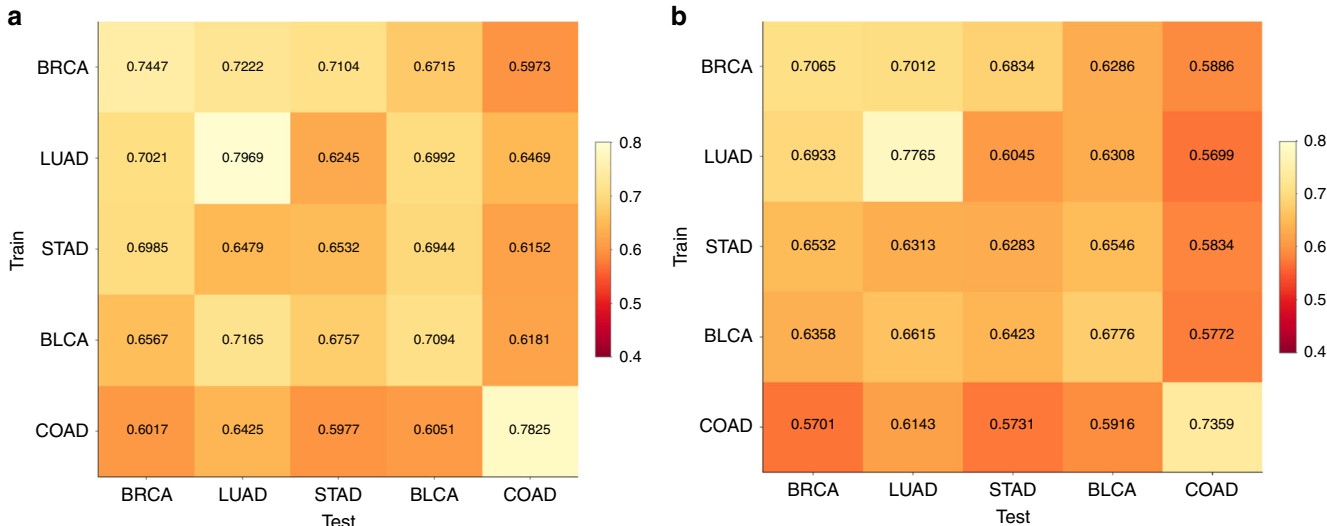

**Fig. 6 AUC of tumor-normal classifiers on the TCGA test set and CPTAC validation set for LUAD and LUSC cancers.** Classification AUCs of each TCGA-trained tumor/normal classifier applied to LUAD and LUSC images from TCGA (reserved "test" data) and CPTAC (external "validation" data, LUAD: n = 1055, LUSC: n = 1060) are shown. **a** Bar graphs comparing test and validation AUCs on LUAD and **b** LUSC slides. **c** Scatter plot of test AUC versus validation AUC for LUAD and **d** LUSC. TCGA train and test sample sizes are provided in Fig. 2a.

**Fig. 7 Classification of TP53 mutation status for TCGA cancer types BRCA, LUAD, BLCA, COAD and STAD.** Cross- and self-classification AUC values from balanced deep learning models (with 95% CIs) are given (**a**) per-slide and (**b**) and per-tile.

sample as cancer. Analogously, the BRCA-trained TP53 mutation status model predicts patterns similar to the LUAD-trained model. Importantly, the tumor/normal and TP53mut/wt classifiers highlight different regions, indicating these classifiers are utilizing distinct spatial features. A caveat to these analyses,

however, is that the spatial variation within heatmaps may reflect TP53mut-associated microenvironmental features rather than genetic variation among cancer cells.

We next performed a tile-level cross-classification analysis as a function of test set. For most test cancer types, we observed little

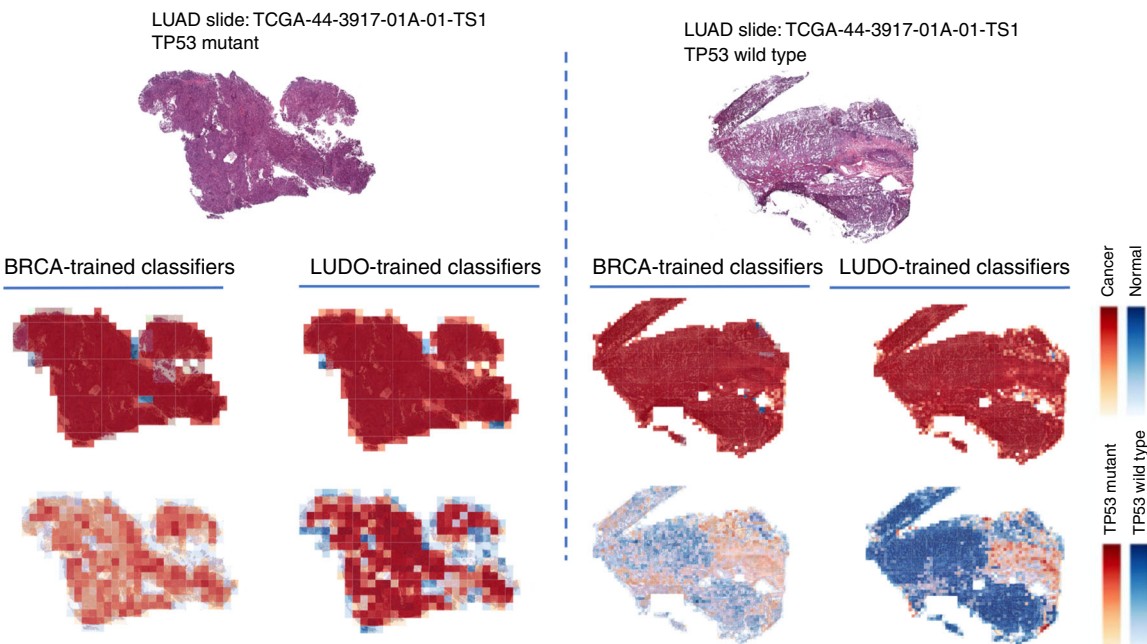

**Fig. 8 TP53 genotype heatmaps based on predicted probabilities using our deep learning model.** The first row shows two LUAD H&E slides with TP53 mutant (left panel) and wild type (right panel). The second row shows prediction maps for these two slides using tumor/normal classifiers trained on BRCA and LUAD samples. Both models successfully classify samples as cancer and predict similar heatmaps. The third row shows prediction maps for these slides using TP53 mutation classifiers trained on BRCA and LUAD. The BRCA-trained and LUAD-trained heatmaps are similar, suggesting that there are spatial features for TP53 mutation that are robust across tumor types.

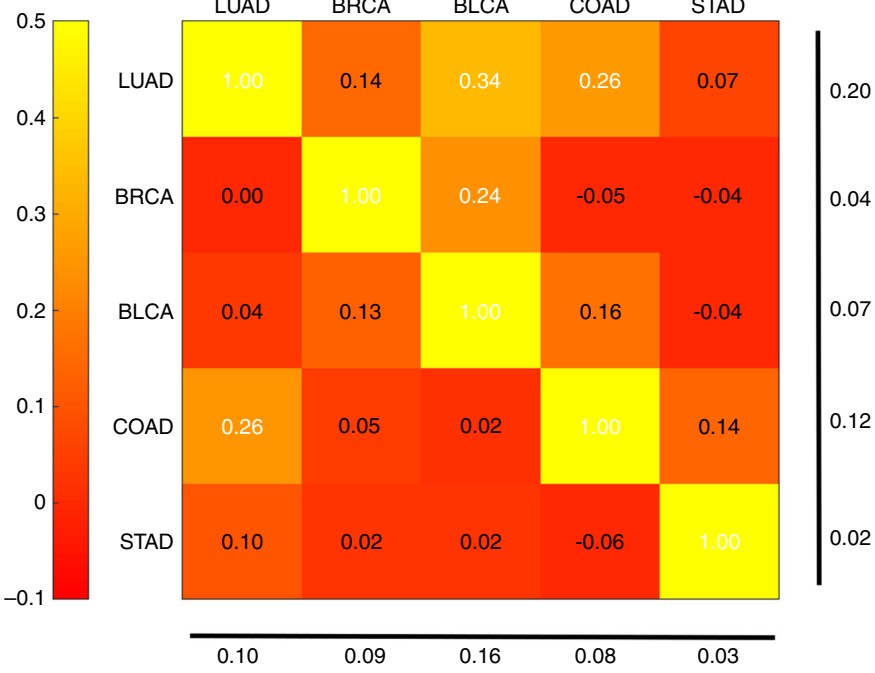

**Fig. 9 Tile-level cross-classification correlations for TP53 mutational status.** Row labels denote the cancer type used to train the first TP53 mutation classifier. Column labels denote both the test tissue and the tissue of the second TP53 classifier. Heatmap values indicate correlation coefficients of mutation probability logits between the two classifiers on the test tissue. Numbers at the bottom and right show column and row averages, respectively, with diagonal values excluded.

correlation when comparing networks trained on cancers "A" and "B" applied to test cancer "C". Therefore, we focused on cases where C is the same as B. Figure 9 plots the correlations of TP53 mutation probability logits across cancer pairs, where each row denotes the cancer type the first CNN is trained on, and each

column is both the test tissue and the second CNN training tissue. In these cases, the correlation coefficients were generally positive and met statistical significance though with moderate magnitude. All correlations were significant, except for the BRCA TP53 classifier applied to LUAD tumors ($t$ test on Fisher z-transformed

correlation coefficients, FDR 5%). Notably, classifiers based on LUAD, BRCA, and COAD worked well on BLCA, BLCA, and COAD tumors, respectively. BLCA and LUAD are the two test cancers with the largest correlations (column average). LUAD and COAD are the two training cancers with the largest correlations (row average). The high row and column averages for LUAD indicate it is canonical both as a test and a training set. Interestingly, the correlations of Fig. 9 are not symmetric. For example, the network trained on LUAD achieves a correlation of 0.34 on BLCA, while the network trained on BLCA has a correlation of 0.04 when tested on LUAD.

**Features impacting tumor purity prediction**. TCGA provides annotations only at the whole-slide level, limiting our ability to build classifiers that resolve predictive features. To better investigate features, we obtained datasets with higher resolution annotations, i.e., BreCaHAD[39] which provides nucleus-level tumor/normal annotations of 162 breast cancer ROIs (see the methods section for details), and 8 colorectal ROIs hand-annotated at nuclear resolution (>18,000 cells) by our group. These annotations provide the ground truth tumor purity (the fraction of tumor cells, aka cellular malignancy) used for the analysis. We then analyzed these data at the tile level ($512 \times 512$ pixels). We trained CNNs vs. the ground truth purity values for each tile, randomly splitting the BreCaHAD dataset into 150 train and 12 test ROIs (a total of >23,000 cells) and using the colorectal set for validation. Purities of the colorectal tiles are spread over a wide range (mean 58%, standard deviation 19.2%), while BreCaHAD purities are higher (mean 87% in the training set), as detailed in Supplementary Fig. 14. These CNNs yielded a mean absolute error of 14% and 15% for the test breast and colorectal sets, respectively. Root mean squared error (RMSE) values were 8% and 20%, respectively. The average prediction for the colorectal datasets (69%) was intermediate between the true colorectal mean (58%) and the breast mean (87%), suggesting that, although the CNN was trained only on breast data, the CNN was able to learn some features common between breast and colorectal tumors. As a comparison, we also calculated RMSEs between purity and TPF as predicted by the TCGA-trained BRCA and COAD classifiers on the colorectal set. These RMSEs were 45% and 39%, respectively. These values were inferior to the BreCaHAD-trained CNNs, indicating that nuclear annotations provide additional predictive information beyond the overall slide label.

We further tested whether purity was being predicted from only the image regions containing individual nuclei, or whether intercellular information was being used. For this, we made use of a CNN classifier[40] that predicts tumor/normal status from individual nucleus images (see "Methods"). We trained on the breast nuclei, and this was able to predict tumor status of reserved breast nuclei images with high accuracy (AUC 93–98%) However, the breast-trained CNN yielded poor predictions on the colorectal nuclei (AUC 56%). We tested whether adding up the contributions of nuclei within each ROI would lead to good predictions at the ROI level. However, the average RMSE across colorectal ROIs was 25%, higher than the RMSE from the tile-based analysis (20%) of the same data. This suggests that, although the tile-based approach is not aware of individual cells, it compensates by using intercellular regions of images.

## Discussion

In this paper, we have presented a versatile CNN-based framework for the pan-cancer analysis of histology images. Using this framework, we were able to train extremely accurate slide-based tumor/normal classifiers in nearly all cancer types, and we also

were able to classify subtypes and TP53 mutation status with significant though less extreme accuracy. Critically, these pan-cancer studies enabled us to compare classifier outputs as a function of training tissue, test tissue, and neural network architecture. Cross-classification for tumor/normal status was successful across most tissues, despite the variations in native tissue morphology and fat content. These studies showed that tumor images have a robust intraslide structure that can be consistently identified across CNN classifiers. Our findings can be viewed within three prongs: identification of pan-cancer morphological similarities, transfer learning as a common feature extractor, and interpreting spatial structures with tumors.

**Identifying pan-cancer morphological similarities**. While other recent works have investigated image-based cancer classification[41,42], cross-classification has until now been little studied. Comparisons of classifiers support the existence of morphological features shared across cancer types, as many cross-cancer predictors achieve high AUCs. While such results are subject to the limitations of the data used, TCGA is one of the largest available sets, and we were able to validate the effectiveness of TCGA-trained classifiers on independent CPTAC lung cancer sets as well. Thus constraints of the TCGA data, such as a lack of FFPE non-tumor images and variable sample quality, do not preclude the development of effective classifiers, though new algorithmic concepts will be essential for individual image analysis questions just as they have been for realizing the value of TCGA sequence data, e.g. refs. [43,44]. Specific cross-classification relationships between types are also informative. Cancers from a common tissue, such as (KIRC, KIRP, KICH), (LUAD, LUAC), and pan-GI cancers are good predictors of each other, and there are also significant similarities within adenocarcinomas and carcinomas, respectively. These findings further demonstrate that cancer tissue of origin and glandular derivation are reflected in image-based cross-classification results. Remarkably, BRCA, BLCA, and UCEC are unexpectedly easy to classify as test sets, showing strong cross-classifiability both at the WSI and tile levels. Further studies are likely to benefit from focusing on these as canonical image types for analysis and method development. Interestingly, this behavior is not symmetric between train and test cancer types. For example, while the network trained on KIRC achieves an AUC > 90% when tested on BLCA, training on BLCA and testing on KIRC results in an AUC < 65%. Overall, the high levels of cross-classifiability suggest that it will be possible to combine images from multiple cancer types to extend and refine training sets. Investigations into the optimal combinations of sets (both positive and negative) may be useful for improving a variety of classification tasks.

A next challenge is to better define the morphological features that underlie cross-classifiability and evaluate their biological relevance. One approach would be to select classifiers with highly similar outputs and then overlap their spatial salience maps[45,46]. Another approach would be to assess if shared morphological features are predictive of shared genomic markers, e.g., via nonlinear canonical correlation analysis (see ref. [47] and [48] for examples). For these types of image morphology questions, careful selection of cancer test sets will be critical. For example, for the TP53 mutation studies we had enough data to identify significant cross-correlations and spatial structures within images, but such analysis will be more challenging for rarer drivers.

**Transfer learning as a common feature extractor**. Transfer learning-based methods take advantage of a universal set of pretrained layers based on non-histological image collections to decompose images into features, an aspect which reduces

computational costs but can potentially limit classification accuracy. While we have observed that transfer-learning networks excel at tumor/normal classification, they have lower accuracy for cancer subtype and TP53 mutation status predictions. This may be because the features associated with subtype and mutation status are not well-represented in the non-histological image collections. For example, Yu et al.[49] reported that TP53 mutation status is associated with the pixel intensity distribution in the cytoplasm and specific texture features within tumor nuclei, and it is possible that such textures are not in ImageNet while tumor/normal classification may be more related to cell shape and size, which are simpler variables more likely to have analogs within ImageNet. We found that fully trained models, which learn all network parameters directly from the cancer images and are computationally more demanding, yielded higher AUCs. Thus the suitability of transfer learning is task-specific, though determining which tasks are suitable is an open challenge. That being said, the ability of transfer-learning-based models to classify tumors remains noteworthy. Even though the Inception network never used pathology images in pre-training, the large set of image-net pre-training images across diverse object classes still led to pretrained feature representations encoding information salient across cancer types. Further incorporation of histopathological sets during pre-training may improve the resolution of classes with more subtle differences, such as those that differ by single mutations, and this will be an important topic for future study. Continued development of transfer-learning methods for biomedical image analysis Raghu et al.[50] and investigations into the general ability of effective representations to encode information for various tasks, as has been discussed in detail by Bengio et al.[51], will both be pertinent.

The effectiveness of CNN architectures can also be impacted by other issues. As the feature representation of the CNNs using transfer learning is optimized for the ImageNet dataset, additional dense layers are necessary when analyzing H&E slides. Although we found that the architecture in Fig. 1a achieves slightly higher AUCs than the original inception architecture without dense layers, the optimal architecture of the dense layers is an open research question. A second issue is class imbalances in the histopathology samples, which can be further exacerbated by intratumoral heterogeneity. For example, we train CNNs by associating all tiles within a slide with the same label, even though tumor slides will contain some regions that are non-tumor. Our approaches still work because classifiers can tolerate some error in the training data[52]. In the machine learning literature, this corresponds to the general problem of multi-label, multi-instance supervised learning with imbalanced data, an active area of research including for medical image data[28,53–55].

**Interpreting spatial structures within tumors.** Self- and cross-comparisons of classifiers can highlight robust spatial structures within tumor images (e.g., Fig. 8), but interpretation remains a major challenge. Neural networks provide only indirect information about the features responsible for such structures, and expert manual pathological analysis of such cases will be essential. Manual analyses may also clarify the identity of predictive features whose existence is supported by CNNs. For example, our comparison of tile and nucleus-level approaches indicated that intercellular regions are useful in predicting tumor purity, but it is uncertain what specific features mediate this relationship. It is worth noting that such analyses would not be possible without mixtures of tumor and normal regions together within images. Thus it will be important to analyze regions with spatial diversity rather than only regions of high purity, which has been the focus of some recent works[41]. Finally, to improve tile-level

understanding using these approaches, further fine-grained pathological annotations with concomitant hypothesis development from the community are vital, e.g., through extended curation of TCGA and other sets. Prior single histology studies have distinguished spatially important regions by training on detailed annotations from pathologists[56], and expansion across histologies would enable further understanding through cross-comparisons of classifiers. Such comparisons of genetically and phenotypically diverse tissues will be a potent approach to reveal morphological structures underlying cancer biology.

## Methods

### Transfer learning

*Sample selection for tumor/normal classification.* Since there are very few normal FFPE WSIs on TCGA, we only considered flash-frozen samples (with barcodes ending with BS, MS, or TS). We selected 19 TCGA cancer types that had at least 25 normal samples. Here we use cancer abbreviations from TCGA as available at https://gdc.cancer.gov/resources-tcga-users/tcga-code-tables/tcga-study-abbreviations.

In total, 24% of the frozen slides are labeled normal (non-tumor) and the remaining are tumors. For each tumor type, the annotated purity values (TCGA "average percentage of tumor cells") span a broad range of values (Supplementary Fig. 2). The samples were randomly divided into 70% training and 30% testing. Stratified sampling was used to balance the ratio of positives and negatives into train and test sets. Here, the term "normal" is used to refer to the adjacent normal slides of a tumor. These slides do not always represent the truly normal tissue, and a more appropriate term might be "non-tumor". Here, we use the "normal" label for convenience, and to remind that the slides are the adjacent normal cuts.

*Sample selection for histopathological subtype classification.* WSI images from ten tissue types were used for subtype classification. FFPE and flash-frozen samples are approximately balanced among the tumor WSIs; hence we used both for subtype classification. The samples were randomly divided into 70% training and 30% testing. Some cancer tissues had subtypes that were available as individual sets within TCGA. These three tissues were LUAD/LUSC (lung); KICH/KIRC/KIRP (kidney); and UCS/UCEC (uterine). For all other tissues, TCGA provided single sets that spanned multiple histopathological subtypes designated by pathologist annotations. This information was available in the TCGA website as "clinical" supplementary files (with filenames formatted as "nationwidechildrens.org_clinical_patient_{CANCERTYPE}.txt)". Only histopathological subtype annotations with at least 15 samples were considered. Samples with ambiguous or uninformative annotations were not included.

The following subtypes were used for cancer subtype classification: brain (oligoastrocytoma, oligodendroglioma, astrocytoma), breast (mucinous, mixed, lobular, ductal), cervix (adenocarcinoma, squamous cell carcinoma), esophagus (adenocarcinoma, squamous cell carcinoma), kidney (chromophobe, clear cell, papillary), lung (adenocarcinoma, squamous cell carcinoma), sarcoma (MFS: myxofibrosarcoma, UPS: undifferentiated pleomorphic sarcoma, DDLS: dedifferentiated liposarcoma, LMS: leiomyosarcomas), stomach (diffuse, intestinal), testis (non-seminoma, seminoma), thyroid (tall, follicular, classical), uterine (carcinoma, carcinosarcoma). Note that the subtype analysis requires only tumor tissue, so it includes some cancers that were not included in the tumor/normal analysis due to minimum data requirements on the normal samples.

*CNN architecture and training.* We used a Google Inception v3-based architecture for pan-cancer tumor/normal classification of TCGA H&E slides. Our CNN architecture uses transfer learning on the Inception module with a modified last layer to perform the classification task. For predicting mutational status, we utilize the same architecture as in Coudray et al.[19] and fully trained the model on TCGA WSIs.

The output of the last fully connected layer of Inception v3 (with 2048 neurons) was fed into a fully connected layer with 1024 neurons. The output was encoded as a one-hot-encoded vector. A softmax function was utilized to generate class probabilities. Each training simulation was run for 2000 steps in batches of 512 samples, with 20% dropout. Mini-batch gradient descent was performed using Adam optimizer[57]. To mitigate the effects of label imbalance in tumor/normal classification, undersampling was performed during training by rejecting inputs from the larger class according to class imbalances, such that, on average, the CNN receives an equal number of tumor and normal tiles as input. Per-tile classification ROCs were calculated based on thresholding softmax probabilities. To compute per-slide classification ROCs, each tile is associated with the class having the highest softmax probability. Then the fraction of tiles labeled as tumor, i.e., TPF (tumor-predicted fraction), is used to distinguish classes at the slide level. Due to significant additional compute costs, we did not optimize on hyperparameters, e.g., number of epochs or learning rate, instead of using common values for similar image classification problems. Details are in the Github code repository.

Preprocessing and transfer-learning steps:

1. Aperio SVS files from primary solid tumors or solid tissue normal samples with ×20 or ×40 magnification were selected.
2. Each SVS file was randomly assigned to train or test set.
3. ×40 images were resized to ×20.
4. The background was removed as in ref. [19]. This step removes regions without tissue and also limits regions with excess fat.
5. Images were tiled into non-overlapping patches of $512 \times 512$ pixels.
6. Tiles were used as inputs of the Inception v3 network (pretrained on ImageNet; downloaded from http://download.tensorflow.org/models/image/imagenet/inception-2015-12-05.tgz), in a forward pass and the values of last fully connected layer ("pool_3/_reshape:0") were stored as "caches" (vectors of 2048 floating-point values).
7. Caches from the same holdout group were shuffled and assigned to TFRecords in groups of 10,000. The TFRecord file format is a simple format for storing a sequence of binary records. TFRecord is TensorFlow's native storage format and enables high data throughput that results in a more efficient model training pipeline.
8. TFRecords were then used as input to the transfer-learning layers.

*Programming details.* All analysis was performed in Python. Neural network codes were written in TensorFlow[58]. Images were analyzed using OpenSlide[59]. Classification metrics were calculated using Scikit-learn[60]. All transfer-learning analyses including preprocessing were performed on the Google Cloud Platform (GCP). The following GCP services were used in our analysis: Kubernetes, Datastore, Cloud Storage, and Pub/Sub. During the preprocessing steps, we used up to 1000 compute instances (each 8 vCPUs and 52GB memory) and up to 4000 Kubernetes pods. Cloud Storage was used as a shared storage system, while Pub/Sub asynchronous messaging service in conjunction with Datastore was used for task distribution and job monitoring of the Kubernetes cluster. This architecture ensures scalability and a fault-tolerant process. We leveraged a similar architecture for the pan-cancer training/testing process.

*External validation using CPTAC images.* The LUAD and LUSC slides of the CPTAC project were obtained through The Cancer Imaging Archive (TCIA) portal. CPTAC denotes lung squamous cell carcinoma by TSCC while TCGA uses LUSC. Here, the term LUSC is used throughout for consistency. Four CPTAC slides in common with TCGA were removed. Slides were tiled and pre-processed similar to the TCGA data, and those with less than 50 tiles were removed. The tumor-normal classifiers were trained on the TCGA data for validation using the hyperparameters described in the "CNN architecture and training" section. The external validation was performed on 1055 CPTAC LUAD (377 normal, 678 tumor) and 1060 CPTAC-LUSC (372 normal, 688 tumor) WSIs.

### Mutational classification

*Sample selection for mutational classification.* We selected flash-frozen WSIs of BRCA, LUAD, and STAD cancer types. Impactful TP53 mutations were determined using masked somatic mutations maf files called by MuTect2[61]. We first considered all called mutations categorized as MODERATE/HIGH (by VEP software[62]) in the IMPACT column. If the gene had at least one such mutation in the sample, it was counted as mutated and was considered as wild type otherwise. Table 1 shows the number of wild-type and mutated slides in each cancer type. For cross-classification, the model was trained on the entire training set and predictions were made on the entire test set.

*CNN architecture and training.* We utilized the Inception v3 architecture[19] to predict TP53-associated mutations in BRCA, LUAD, and STAD sets. Unlike the tumor/normal analysis, transfer learning was not used for mutational classifiers. Instead, models were fully trained on input slides. As a preprocessing step, we used a fully trained normal/tumor classifier to identify and exclude normal tiles within each tumor slide. This filtering step ensures that tiles with positive mutation class label are also labeled as tumor. To predict mutations in the TP53 gene, we trained two-way classifiers, assigning 70% of the images in each tissue to training and the remaining 30% to the test set. The cross-tissue mutational classification was performed by training the model on the entire train set of a cancer type and performing prediction on other cancer types. The model outputs for tiles were used to produce slide-level prediction by averaging probabilities. Similar downsampling as in the tumor/normal classifier was performed to handle data imbalance issues.

*Computational configuration.* All of the computational tasks for mutation prediction were performed on linux High performance computing clusters with the following specification: 8 CPUs, RAM: 64 GB, and Tesla V100 GPUs, 256 GB RAM. Furthermore, The GPU-supported TensorFlow needed CUDA 8.0 Toolkit and cuDNN v5.1. All GPU settings and details were obtained from TensorFlow and TF-slim documentations and NVIDIA GPUs support.

*Cross-classification statistics.* Hierarchical clustering was applied to cross-classification per-slide AUC values using UPGMA with Euclidean distance. To determine the association between clustering and independent phenotypic labels (i.e., organ and adeno-ness), we used Gamma index of spatial autocorrelation from the Python package PySal[63]. Gamma index is defined as[64]:

$$\Gamma = \sum_{i,j} A_{ij} W_{i,j}, \tag{1}$$

where $A$ is the feature matrix and $W$ is the weight matrix, and indices range over cancer types. For each axis and each phenotype group (i.e., organ or adeno-ness), we calculate a separate Gamma index. We define $A_{i,j} = 1$, if cancer types $i$ and $j$ have the same phenotype (e.g both are adenocarcinoma) and $A_{i,j} = 0$ otherwise. For weights, we set $W_{i,j} = 1$ if cancer types $i$ and $j$ are immediately clustered next to each other and $W_{i,j} = 0$ otherwise. $P$ values are then calculated by a permutation test using the PySal package. We dropped any cancer type with "Other" phenotype from this analysis. To avoid extensive computation cost for computing CIs, we used the method of Reiser[65] to compute CIs instead of generating bootstrap sub-samples. A similar procedure is used to compute the CIs of tumor/normal and cancer subtype classifiers. In order to compute tile-level correlations, we first compute tumor probability logit for each tile, defined as $\log((p + \varepsilon)/(1-p + \varepsilon))$, where $P$ is tumor probability and $\varepsilon = 0.0001$ is added to avoid dividing by or taking the logarithm of zero.

*Patient-level stratification.* We considered two ways of splitting data by patient for the analysis of Supplementary Fig. 1. (1) First, we considered two patient groups—those with adjacent normals and those without. For each cancer type, 70% of patients in each group were randomly assigned for training, and the remaining 30% were used for testing. Slides corresponding to each patient, whether in train or test, were placed in their associated class, i.e., normal or tumor. This data split was denoted by the "patient level" stratification. (2) Alternatively, we restricted analysis to patients who only have adjacent normal and used the 70/30 split of patients. This split was denoted by "matched patient level" stratification.

### Purity estimation for BreCaHAD and colorectal nuclear annotations

*BreCaHAD dataset.* The images of this dataset are based on archived surgical pathology example cases used for teaching purposes[39]. "All specimens were breast tissue fixed in 10% neutral buffered formalin (pH 7.4) for 12 h, processed in graded ethanol/xylene to Surgiplast paraffin. All sections were cut at 4-micron thickness, deparaffinized and stained with Harris' hematoxylin and 1% eosin as per standard procedures"[39]. This dataset contains 162 ROIs, where each ROI is 1360-by-1204 pixels and is obtained at the ×40 magnification. The data is saved using the uncompressed ".TIFF" format.

*Tile-based purity estimation.* We used Inception, DenseNet, and Xception-based transfer-learning models, each trained for 20 epochs, where the network at the epoch >10 performing best on test data and having test mean squared error larger than the train set is used for validation. Tiles of size $128 \times 128$ resulted in large test errors, and the tiles were too small for tiles of $1024 \times 1024$ tiles. We, therefore, focused on tiles of sizes 512 and 256, and tiles of size 512 for validation. For cases with reduced magnification, we downsampled 512-by-512 tiles by a factor of two. To correct for acquisition differences between breast and colon cancer ROIs, we equalized the tile histogram distribution. For each tile, purity is defined as the ratio of tumor cells to total cells. We slightly adjusted to avoid purities too close to zero or one, as these may destabilize the analysis, i.e., given a tile with purity value $P$, we compute logit purity as $\log((P + 0.05)/(1.05-P))$, then invert the logit to obtain adjusted purity values. We used overlapping tiles with step size 64 pixels for both tile sizes. Given the extracted features, we used a fully connected layer of 256 neurons with ReLU non-linearity, followed by a dropout of 25%, and a fully connected neuron using the sigmoid activation. We used the "he_normal" initialization method of Keras described in He et al.[66], and an elastic net regularization setting L1 and L2 penalties

**Table 1 Numbers of wild-type and mutated slides in each TP53 cancer set.**

| Cancer type | No. of wild-type slides | No. of mutated slides | No. of train slides | No. of train tiles | No. of test slides | No. of test tiles |
|---|---|---|---|---|---|---|
| BRCA | 647 | 338 | 699 | 438,813 | 286 | 198,580 |
| LUAD | 295 | 270 | 396 | 452,419 | 169 | 193,245 |
| STAD | 237 | 200 | 306 | 428,872 | 131 | 176,059 |
| BLCA | 217 | 194 | 276 | 125,003 | 112 | 60,061 |
| COAD | 184 | 214 | 283 | 150,881 | 115 | 60,312 |

to 1e-4 (the penalty value performing best on the test set compared with two other penalties: 1e-3 and 1e-5).

*Nucleus-based purity estimation*. We implemented the network of Hlavcheva et al.[40], including their reported hyperparameters. The goal of this method is to classify individual nuclei as tumor or normal. Nucleus patches were resized to 32-by-32 pixels. To adjust for acquisition differences between the breast and colon datasets, we applied histogram equalization to both datasets. We trained on the BreCaHAD training set and tested on the reserved breast data across all individual nuclei, finding high accuracy (AUC 93–96%). For comparison, we also tested a transfer-learning approach. The transfer-learning pipeline used similar preprocessing, except nucleus patches were resized to 128-by-128 pixels since Inception requires images to be larger than 75-by-75 pixels. The fully trained method was superior to transfer learning (all transfer-learning AUCs <65%, over various parameter choices). Therefore for analysis of the colon cancer dataset, we used the Hlavcheva et al fully trained method, trained on the entire BreCaHAD dataset. For predictions of TPF on ROIs, we compared the sum of predicted tumor probabilities across all nuclei to the pathologist purity annotations of all cells in the ROI.

**Reporting summary**. Further information on research design is available in the Nature Research Reporting Summary linked to this article.

## Data availability

The TCGA data can be downloaded from genomic data commons at https://portal.gdc.cancer.gov/. The CPTAC data can be downloaded from the cancer imaging archive at https://www.cancerimagingarchive.net/. The BreCaHAD dataset can be downloaded from https://figshare.com/articles/BreCaHAD_A_Dataset_for_Breast_Cancer_Histopathological_Annotation_and_Diagnosis/7379186. Additional data used in the study can be found at the github page https://github.com/javadnoorb/HistCNN.

## Code availability

Code used in this analysis can be found on the GitHub page: https://github.com/javadnoorb/HistCNN. A variety of public datasets have been used as the basis for these studies. Details for their access can be found in "Methods", with additional information on the GitHub page.

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

## Acknowledgements

This research benefited from the use of credits from the National Institutes of Health (NIH) Cloud Credits Model Pilot, a component of the NIH Big Data to Knowledge (BD2K) program. This material is based upon work supported by Google Cloud. J.H.C. acknowledges support from NCI grant R01CA230031. J.N. and J.H.C. would like to thank Yun-Suhk Suh for helpful comments.

## Author contributions

J.N. and S.F. developed the TCGA CNN implementations, analyzed the TCGA data, and drafted the paper. A.F. developed and analyzed the CNN implementations for the nuclear data, contributed to the analysis of TCGA data, and drafted the paper. D.C. and D.R. developed the nuclear cell datasets, contributed to data analysis, and provided pathological evaluations. M.S. developed the cloud software engineering approaches and contributed to data analysis and drafting of the paper. K.Z. oversaw the methods development and statistical analysis and drafted the paper. J.H.C. led the project and finalized the paper.

## Competing interests

D.L.R. has served as an advisor for Amgen, Astra Zeneca, BMS, Cell Signaling Technology, Cepheid, Daiichi Sankyo, Danaher, GSK, Konica/Minolta, Merck, Nanostring, NextCure, Odonate, Perkin Elmer, PAIGE.AI, Roche, Sanofi, Ventana, and Ultivue, and he has equity in PixelGear and receives royalties from Rarecyte. He has received instrument support from Ventana, Akoya/Perkin Elmer, and Nanostring. Amgen, Cepheid, Navigate BioPharma, NextCure, Konica/Minolta, Lilly, and Ultivue fund research in D.L.R.'s lab. The remaining authors declare no competing interests.
