## [Peer Review File · Nature Communications]

Reviewers' Comments:

Reviewer #1:

Remarks to the Author:

This report focuses on the analysis of whole slide images from frozen and FFPE H&E whole slide images (WSI) from 19 tumour types of TCGA. CNN are used to train detection of tumour and normal from a tiling approach. Training and test sets are used and comparisons of models across tumour types is performed for estimation of tumour purity, diagnosis, tumour subtype, and TP53 mutational status. Comparisons of how models trained in one tumour type perform across other tumours is rigorously assessed and presented. It is the true novelty of this work and of considerable interest. Correlations and clustering are presented for broad classes such as adenocarcinoma versus carcinoma versus other as well as organ of origin (GI, kidney, GYN, lung). I have a few comments.

This paper is driven more by analytic models than biology and practical pathology considerations and some aspects are not clear to me. I list points to consider for clarification below:

- (1) Frozen and FFPE slides have very different properties, yet this is not discussed and performance of models between these two data sources is not presented.
- (2) In terms of training, for frozen, slides designated as tumour and normal are relatively pure and homogenous within the guidelines for acceptance as a curated TCGA sample, though level of neoplastic content vary with PAAD being notoriously low and most others being uniformly high. However, for FFPE, these were representative slides for each case and many times contained mixtures of tumour and normal. Did this affect model building? Also, how much did the differences between Frozen and FFPE samples affect model performance?
- (3) Estimations of tumour purity versus TCGA reference pathologist assessment for frozen section (this was not performed in TCGA for the FFPE images) were quite low for STAD, LIOHC, PAAD, ESCA and KICH. Is there a reasonable explanation for this?
- (4) In Figure 4, clustering is examined for various malignancies based on the identity of adenocarcinoma versus carcinoma versus other. The definition of adenocarcinoma is a bit squishy when examined closely -- functionally these are derived from lineages that perform secretory functions within organs. Operationally, one looks for features such as gland formation or intracellular mucin to define this status in pathology slides. The authors seem to have defined tumour as carcinoma versus adenocarcinoma based on their TCGA name. These TCGA names are deceptive and not always associated with lineage status. For instance, BRCA is definitely an adenocarcinoma, as is UCEC and others. This needs to be looked at much more closely with some reclassification. The authors do rightly glean that ESCA is a mixture of squamous and adeno cases and thus designate it as other along with sarcoma which is a mesenchymal family of tumours.
- (5) More focus on the tumour biology implications of this work with suggestions of why relations might be seen or expected would be welcome in the Discussion section.

I have no further comments. Thanks for the chance to review this very interesting work.

Reviewer #2:

Remarks to the Author:

This manuscript describes an analysis of 27,815 images derived from 19 different cancer types from TCGA using a convoluted neural network (CNN) approach to distinguish cancer vs. normal, achieving an AUC=0.995, and discriminate specific subtypes (variously defined) with an AUC=0.87. CNNs trained on one tissue were effective in distinguishing cancer vs. normal in test

sets derived from cancers not included in the training set. Highest levels of discrimination were achieved for breast, bladder and uterine cancers (versus normal). The CNN achieved discrimination for TP53 mutant cancers from wild type (AUC=0.65-0.80). The manuscript concludes that the universality of performance of the CNN across different tumor types suggests a capacity to perform "cross-comparisons to reveal conserved spatial biology". The manuscript attempts to address under-explored areas of potential interest: 1) how CNNs trained on one cancer type perform on other cancer types; 2) the efficacy of transfer learning and 3) factors that predict classification accuracy. The analysis also addresses how cellular versus inter-cellular tissue regions impact classification performance. This is an ambitious project with substantial potential; however, the breadth of the work comes at expense of depth, which limits interpretation of the data. At this stage in the development of CNN approaches, exhaustive interactions with "pathologists" (which can be non-pathologists with relevant expertise) is generally required to turn data into knowledge that can be leveraged to improve biological understanding or achieve clinical translation. This project includes world class senior pathologists, but would benefit from interactions with pathologists who have the time to work closely with the images and data. Perhaps most importantly, it is useful in such analyses to assess independent and meaningful gold standards, which are not related to classification of training and test sets. The specific analysis of TP53 status is a good example, but there are innumerable others that could have been considered, and this would assess whether the CNN provides information that microscopists cannot. The use of TCGA in this experiment is questionable. TCGA is a skewed collection of cancers that were identified to develop a working molecular analysis of all cancer types. Studies that leverage these strengths make best use of the resource (analysis of pan-omics data for hypothesis generating). The current analysis focuses mainly on comparing a CNN classifier to pathologists' interpretations of morphology and representativeness of pathology is not the strength of TCGA.

Major Points

1. The source of TCGA image sets and the features of these images are not adequately described. For example, TCGA was restricted to larger tumors that could be collected fresh and snap frozen for molecular analysis. TCGA samples were excluded if percent tumor composition did not meet specific thresholds. Further, image quality may have been an issue, specifically for frozen sections; although such images are of generally reduced quality compared to sections of unfrozen tissue, nearly all frozen tissues are made from cancers, and the artifacts of freezing could trigger the correct classification of cancer by the CNN. There are likely many issues related to size, quality and preparation of images that could affect CNN performance, and impact either the accuracy or generalizability of the results. No analysis is perfect, but knowledge of these factors is helpful in putting the results into perspective.
2. What was the source of the images of "normal"? This is an incredibly important issue, with substantial implications for interpretation of the data. How were normal samples selected and what can we say about the composition of these images? For example, normal breast age-matched to breast cancer is largely fat and contains few epithelial cells, whereas cancer tissues are highly cellular. Thus, what appears to represent accurate discrimination of breast cancer from normal breast tissue could be driven partly (or largely), but a simple assessment of cell density, which is not a great achievement for image analysis with or without the application of a CNN. Similar arguments would apply at the level of a slide or an image tile within a slide. Using "normal" TCGA tissues is not ideal because these are normal tissues associated with cancers. Data show that such tissues are often not normal, using various approaches, ranging from morphologic to molecular.
3. Without knowledge of the composition of various test sets, the interpretation of AUC is unclear. While true that the sensitivity and specificity of test performance are independent of the case vs. control percentages of a sample, what is more important is the predictive value of a classifier. For example, even a high specificity would give a poor positive predictive value if the prevalence of the disease is low.
 - a. These considerations could also impact the assertion that certain cancer types are "canonical" examples of the power of CNNs to offer important classifications. We do not know the composition of cancer vs. normal per tumor type or have a sense of the composition of tumors and normal samples. In the uterus, the definition of "normal" is as troubling as for breast; normal could mean atrophic, proliferative, secretory or something else and each of these morphologic appearances is

quite distinct with regard, cellularity, cell organization, cytology, etc.

4. The section of tumor-predicted fraction and tumor purity requires explicit definitions of each term. What is the gold standard and why should we accept that the proposed definition is the most accurate metric? This is a complicated area, and optimal metrics may vary by applications. If the experiment is DNA based, then knowing percent DNA purity is critical (cancer vs. not cancer), whereas if the issue related to percentage of cells demonstrating a particular marker (e.g. immunostain), then percent of cells that is tumor is critical.

5. The conclusion that cancer vs. non-cancer is a fundamental distinction that can be achieved by training a CNN is perhaps intuitive. These "classes" are vastly different with regard to nearly any feature: morphological, molecular, cellularity, etc. Demonstrating the universality of cancer features across tumor subtypes is of some interest, but it would be useful to provide greater insights into the drivers of this distinction if possible. What are the features of the cancers misclassified? Were they reviewed by a pathologist?

a. The TCGA set is peculiar and not ideal for making generalizable conclusions about many issues in pathology and biology. However, it may be useful for hypothesis generating. Over-fitting is a major problem in developing CNNs, and the peculiarity of TCGA samples weighs on this issue. It would be much stronger to test the CNNs on non-TCGA samples to accurately assess performance. Even if this were done in a limited way, it would help understand these data.

6. Interpretation of data on a per tile basis may be limited by lack of description of the size of tissues imaged, magnifications, and purity, etc.

7. The ability of the CNN to discriminate cancer with TP53 mutations is of interest, but the analysis does not provide depth of understanding. How does this relate to grade or tumor size or cellularity? Has anyone visually compared these tumors in relation to CNN classification? Are there penultimate layers of the CNN that can provide analytical data that could offer insights (morphometric features)?

Minor Points

1. Philosophical question: Is the goal of the current project to demonstrate how CNNs can reduce human effort, as implied by the introductory statement, "Manual inspection is time consuming ..." or to demonstrate the potential for computers to extend the horizons of morphological characterization of tissues or both?

2. The Abstract refers to "slides" but the analysis was performed on scanned images, a non-trivial distinction.

3. The Abstract refers to "accuracy" but the gold standard of accuracy must be stated.

4. The Abstract conclusion that CNNs may perform "cross-comparisons to reveal conserved spatial biology" is rather vague or obscure, without specific meaning. A more tangible, if somewhat unglamorous sounding statement, would be more meaningful. CNNs can distinguish cancer from non-cancer based on their potential to appraise features used in histopathologic diagnosis, such as cellular density and organization (if this is what is meant).

5. The section "Neural network classification of cancer subtypes" is confusing.

a. Some cancers list subtypes that are chosen for discrimination, such as lung adenocarcinoma vs. squamous cell carcinoma, but others are not (e.g. sarcoma). The text and figures would benefit from clarification and revision to increase consistency and clarity.

b. Lists of cancer subtypes included in analyses might be relegated to supplement.

c. Reference to cervix and adenoma, presumably should read adenocarcinoma.

6. The manuscript includes several statements in which performance of the CNN as a classifier is invoked as a means of providing biological insights, but such statements are not specified. What sorts of biological insights are provided and do these expand our knowledge of cancer biology?

7. Use of the word "cohort" is unfortunate. TCGA is not so much a cohort of any sort, but rather a tissue collection with extremely modest annotation of patients and clinical factors. Indeed this is a major limitation of TCGA.

8. A better description of "transfer learning" would increase understanding for readers less familiar with CNN as applied to morphology.

9. Table 1 reference to slides in relation to TP53 analysis is not necessarily apt. TCGA data imply that somewhere in the tumor (the part that was ground up and tested) a mutation was or was not found. Unfortunately, we do not know for sure if there was a mutation in the part of the tumor that

was used to create the image of the morphology do we?

Reviewer #3:

Remarks to the Author:

The authors evaluate the role of deep learning using digital images of tumors from different TCGA cohorts and indicate that a convoluted neural network (CNN) training in one tumor type can recognize and accurately classify tumors from normal tissue and TP53 mutation status in a different organ-type cohort. The authors conclude that this cross classification has biological relevance.

The article is difficult to understand and read as there are many abbreviations for tumor types used that are not clearly indicated in the text.

The authors use many different tumor types including sarcomas and brain tumors, but concentrate their efforts throughout the article on epithelial tumors (carcinomas). It would be better to report only work on carcinomas, as the non-carcinoma types do not perform well in their model.

Fig 2: Not all tumor types indicated are described in the text. The text indicates that testicular tumors were included, but they are not represented in the figure. Testis is indicated in Fig 3, but other tumor types are missing from that figure.

Fig 4: similar issues with abbreviations and tumor type. In addition, the authors suggest that clustering of lung and kidney; and breast and GYN tumors. However, there is no further explanation. Was the clustering related to specific tumor types, eg. endometrioid adenocarcinoma of the uterus versus colon cancer? Lung cancer and kidney cancer do not seem to share morphological similarities. Could nuclear hyperchromasia be responsible for the clustering?

Can these findings be validated in a non-TCGA cohort?

Reviewer #4:

Remarks to the Author:

The authors propose a deep learning network for binary and multi-class classifier for 19 different cancer types. A large number of experiments on a large number of multi-tissue TCGA dataset are reported. Using the standard Inception-v3 network for both binary and multi-class classification of TCGA cases into tumor and normal and subtyping of tumors, the authors report AUC scores of 0.995 ± 0.008 and 0.87 ± 0.1 for binary and multi-class classification, respectively. They further reported results of cross-tissue type performance analysis of their classifiers again showing results in AUC score of 0.88 ± 0.11 . They conclude that some cancer types are relatively easy to classify as compared to others; TP53 mutation can be predicted from histology images with AUC score range 0.65-0.80.

I have following concerns regarding the manuscript:

- In Binary classification, all tiles of the normal case are considered normal and all tiles of the malignant case are considered as tumors. Given a tumor sample may contain a large number of normal tiles and only a handful of tumor tiles, how can that approach be justified?
- It is stated that the tumor-predicted fraction (TPF) in each slide is then "used as a metric". It is stated that per-tile classification ROCs were calculated based on thresholding softmax probabilities and per-slide classification ROCs were based on voting on maximum softmax probability. Again, a slide can be labeled as a tumor slide even though a very small number of tiles (even only one tile) is labeled as tumor. How would that work?
- A standard Inception v3 network trained on ImageNet is used with a basic transfer learning method that trains the final couple of layers on the histology image data for both binary and multi-

class classification. Although, high AUC score values support the claim that transfer learning work, a discussion of how a network trained on natural images work on histology images, when the contents of two types of images are quite different, is missing.

- For the tumor/normal classification CNN, all training data is based on frozen slides and results in AUC of 0.995, whereas in cancer subtype classification both FFPE and Flash Frozen are used, the AUC is decreased to 0.87. A discussion of the drop in performance is missing.

- For TP53 mutation, AUC scores below 0.75 could not be considered reasonable. What is the F1 score like? If that's below 0.7, a histology based test would be of no use.

- Deep learning-based approaches usually require a validation set along with training set for hyperparameter tuning and optimal model selection. Authors have used test set as validation set for "Tile-based purity estimation". However, it is not clear that how they selected the optimal models for all other experiments.

- Authors listed "transfer learning as common feature extractor" as one of their findings which is already well studied topic and number of methods have used transfer learning based features in multiple image modalities.

- Since frozen slides alter the tissue appearance drastically, therefore, in practice, these are not considered suitable for computational analysis. Therefore, a separate experiment with only FFPE based training and testing of the pipelines is highly recommended.

- Similarly, the cross-classification between tumor types is recommended to be evaluated with FFPE samples only. This is important to establish that the flash-frozen special appearance and resulting tissue damage is not the dominant factor in discriminating normal vs tumor slides and so on.

- It seems that for every cohort, some of the images are used for training and the remaining for testing. A true cross-classification experiment would be leave-one-TCGA-cohort-out and reporting average results on those experiments. It is recommended to run cross-validation experiments for each classification task.

- TP53 prediction is slightly more accurate using self-cohort than cross-classification, which is claimed to be the main contribution of the paper.

Minor comments:

- Many cases in the TCGA cohort contains multiple slides, therefore, all slides of a case should be part of only one dataset/data-split either train or test. However, it is not clear from the text that whether authors have followed this protocol or not.

- Figure 1a: it is quite difficult to infer the number of slides due to log-based scale. Therefore, bar-chart based visualization would be better.

- The step of TFRecord conversion (Figure 1a and 1b) in the deep learning pipeline is not well-understood, what is it and why it is introduced in the pipeline is not well explained, is it just an efficient implementation or if it has a role in the network prediction too? Though, it seems core TensorFlow data structure/type.

Reviewer #5:

Remarks to the Author:

While the study of distinguishing cancer tissues from normal is an interesting problem to test the methodology, it is of low impact, as one can imagine that this would be trivial for pathologists to do. Similarly, several other modalities (e.g. transcriptomics, epigenomics etc.), can be easily used to classify cancer vs. normal. Similarly, "classical" computer vision approaches using morphometric features etc. have been used on WSI image data, to classify cancer vs. normal with the same performance. The same holds for distinguishing cancer subtypes. So how surprising is it that CNNs can do this?

It is not clear if the FFPE & frozen slides can be used together? They have very different potential for artefacts, and the results can easily be biased towards different way of storing and preparing the slide. Is there any data to show that models built on both do not suffer from these potential issues?

Even though there is a statistically significant correlation between TPF and pathologist reported purity, actual correlation is still quite low, most cancers are in the 0.2 vs. 0.4 range. What can be concluded from this analysis? What happens if you correlate with genomic measures of tumor purity?

Couldn't the results in Supplementary Figure S2, just be explained since in TCGA you have many more slides that are tumor vs. normal, which means the same for tiles, and as such, you will overpredict tumor?

Do the authors have any interpretation why the transfer learning results don't work for TP53 mutation status?

It is not clear if you can use from the same patients tiles for training and testing? This would get dangerously close to overfitting, especially as the authors first use their cancer/normal classifier to discard "normal tiles", that means that already a selection is made for tiles that are similar. Can the authors show that results would be similar if using tiles for testing from left out patients?

For the TP53 analysis, did the investigators distinguish between gain and loss of function TP53 mutations? From a functional perspective the models should be able to distinguish the difference between these two types of mutations, and without considering this, there could be a bias in the analysis.

Most of the analyses report AUC and accuracy as performance metrics, however, in the last analysis on the separate breast cancer & colorectal cancer cohort, the authors report RMSE. This makes the results difficult to compare? What are the accuracy results of these classifiers, when using similar model for tumor purity?

Similarly, what happens when you use models trained on TCGA data and test on the new cohorts with local ground truth?

It seems the authors are using different CNN models for each analysis? Does this suggest that a single modeling framework or architecture using CNNs is not able to learn the different tasks?

The results described on lines 198 till 215 are hard to follow including the associated figure, can this analysis be explained in different way/improved/etc. ?

Specific comments:

- Methods Line 413: Can the authors be more specific what they mean with "clinical subtype". This is very vague? Do they refer to histopathological subtype?

- The authors use the Tensorflow specific format TFRecord, this should be properly defined and introduced for the reader that is not familiar. Especially in Figure 1, it would be more appropriate

to use a more general term referring to this step.

- The BreCaHAD data set should be better introduced in the methods section.

Dear Dr. Righetto,

We would like to thank the reviewers and Nature Communications for closely reading our manuscript and for the detailed comments. We have conducted substantial additional analyses as well as improving conceptual descriptions to address the issues brought up by each reviewer. These new aspects include testing on a new external dataset (CPTAC), comparison of frozen and FFPE slides, and further manual pathology analysis, among many others. Please see below for point-by-point responses. In the manuscript, edits in response to reviewers are highlighted in blue. We have also made a few grammatical and wording fixes, though these are minor and have not been tracked.

Best regards,
Jeff Chuang

REVIEWER COMMENTS

Reviewer #1 (Remarks to the Author):

This report focuses on the analysis of whole slide images from frozen and FFPE H&E whole slide images (WSI) from 19 tumour types of TCGA. CNN is used to train detection of tumour and normal from a tiling approach. Training and test sets are used and comparisons of models across tumour types is performed for estimation of tumour purity, diagnosis, tumour subtype, and TP53 mutational status. Comparisons of how models trained in one tumour type perform across other tumours is rigorously assessed and presented. It is the true novelty of this work and of considerable interest. Correlations and clustering are presented for broad classes such as adenocarcinoma versus carcinoma versus other as well as organ of origin (GI, kidney, GYN, lung). I have a few comments.

This paper is driven more by analytic models than biology and practical pathology considerations and some aspects are not clear to me. I list points to consider for clarification below:

(1) Frozen and FFPE slides have very different properties, yet this is not discussed and performance of models between these two data sources is not presented.

The TCGA database has very few (<20) normal FFPE whole slide images. As a result for the tumor vs normal classification task we only used frozen samples. This was not quite clear in our original manuscript. So in this edition we added the following (p.3): "Due to insufficient FFPE normal WSIs in TCGA, for this task we only used flash frozen samples."

However, for the subtype classification task we were able to use both frozen and FFPE samples. The models described in the results were trained on frozen and FFPE images combined together. To address the reviewer's concern, we tested the predictive accuracy of the CNNs on just the frozen samples or just the FFPE samples. The CNNs classified the FFPE and

frozen samples with comparable accuracy, with the same tumor types doing better (e.g. kidney subtypes) or worse (brain subtypes) across the two types of samples (see below). These results indicate that the CNNs make use of features not specific to the sample preparation method.

FFPE sample subtype classification accuracy

Frozen sample subtype classification accuracy

We have added Supplementary Figure S4, which presents these results as well as a correlation analysis across the sample types (p. 33). We have also added the following text to the manuscript (p.5):

“The images used in the subtype analysis were from a mixture of frozen and FFPE samples. Although FFPE samples are preferred because they avoid distortions caused by freezing, we tested whether the CNNs were able to classify subtypes for each sample preparation (Figure S4). The CNNs classified the FFPE and frozen samples with comparable accuracy, with the same tumor types doing better (e.g. kidney) or worse (brain) in each. Correlations between classification AUCs were high across the two sample preparations ($r = 0.87$ for macro-averages; $r=0.78$ for micro-averages). As expected, FFPE-based classifications tended to be better, notably for brain and sarcoma samples.”

(2) In terms of training, for frozen, slides designated as tumour and normal are relatively pure and homogenous within the guidelines for acceptance as a curated TCGA sample, though level of neoplastic content vary with PAAD being notoriously low and most others being uniformly high. However, for FFPE, these were representative slides for each case and many times contained mixtures of tumour and normal. Did this affect model building? Also, how much did the differences between Frozen and FFPE samples affect model performance?

These concerns are addressed in comment 1 above. Unfortunately, subtypes were not available for PAAD to include in the analysis.

(3) Estimations of tumour purity versus TCGA reference pathologist assessment for frozen section (this was not performed in TCGA for the FFPE images) were quite low for STAD, LIOHC, PAAD, ESCA and KICH. Is there a reasonable explanation for this?

We agree with the reviewer that this is an important though complex topic. We believe the underlying reasons include: (1) Due to the weakly supervised nature of the training process, the network may be associating cancer-related image features such as necrosis and lymphocyte infiltration with predicted tumor status, but such features vary in importance as a function of cancer type. The table below lists correlations between TPF (tumor-predicted fraction) and various TCGA slide-level image annotations, the strengths of which vary substantially by tumor type; (2) TCGA assessments of purity have non-negligible uncertainties. For example, TCGA lists the purity of the slide below as 73%, while our team's pathology assessment is that the purity is less than 20% (Our network's TPF is 10%); and (3) Different pathology measures of purity may be more suited to particular cancer types. Our original analyses compared TPF to TCGA values of "average percentage of tumor cells" as the measure of purity; however, the TCGA measure "average percentage of tumor nuclei" yields different correlations. Notably, for ESCA there is a much stronger association of TPF to "average percentage of tumor nuclei" (see table below). LIHC may be a particularly informative cancer for further analysis. For this cancer type, TPF has low correlations to all of the TCGA pathological quantifications. Moreover, most classifiers trained on other cancers have low cross-classification accuracy on LIHC (see Figure 4). Determination of the biological features underlying these TPF estimates is a major endeavor that we are now pursuing in follow-up studies (e.g. using attention-based methods and genetics/CNN associations, manuscript in preparation).

As these explanations are not yet conclusive, we have addressed the issues by adding the following edits.

p.4: "Additionally, pathologist assessments of tumor purity have non-negligible variability (Viray et al. 2013) that may affect correlations."

P.9: "Overall, the high levels of cross-classifiability suggest that it will be possible to combine images from multiple cancer types to extend and refine training sets. Investigations into the optimal combinations of sets (both positive and negative) may be useful for improving a variety of classification tasks."

Figure: A STAD slide (TCGA-FP-8211-01A-01-TS1). TCGA provided purity is 73% while our pathologist's assessment is that purity is less than 20% (Our CNN's TPF is 10%).

Table: Pearson correlation of TPF versus TCGA-annotated pathology features.

cancer	purity (avg percent tumor cells)	purity* (avg percent tumor nuclei)	lymphocyte infiltration	necrosis	stroma
esca	0.085789669	0.394298244	-0.172432465	0.17996741	-0.14148905
kich	0.081703071	-0.021962132	0.189155483	0.119581416	-0.16775387
lihc	0.028023649	-0.093919563	0.051307659	-0.07608796	0.034883372
paad	0.196498514	0.068849387	0.076748197	0.012670186	-0.22724248
stad	0.079627024	0.04477744	0.125179155	-0.059428871	0.116203214

(4) In Figure 4, clustering is examined for various malignancies based on the identity of adenocarcinoma versus carcinoma versus other. The definition of adenocarcinoma is a bit squishy when examined closely -- functionally these are derived from lineages that perform secretory functions within organs. Operationally, one looks for features such as gland formation or intracellular mucin to define this status in pathology slides. The authors seem to have defined tumour as carcinoma versus adenocarcinoma based on their TCGA name. These TCGA names

are deceptive and not always associated with lineage status. For instance, BRCA is definitely an adenocarcinoma, as is UCEC and others. This needs to be looked at much more closely with some reclassification. The authors do rightly glean that ESCA is a mixture of squamous and adeno cases and thus designate it as other along with sarcoma which is a mesenchymal family of tumours.

We thank the reviewer for pointing out this issue. The labels of Figure 4 have been proofed and slightly updated by our pathology team. The statistical analysis has been redone based on the updated labels with the conclusions remaining similar. We edited the text accordingly:

P.6: "The inter-cancer distances were significantly associated with adeno-ness on the train axis (p -value=0.015)."

(5) More focus on the tumour biology implications of this work with suggestions of why relations might be seen or expected would be welcome in the Discussion section.

This type of large-scale cross-classification analysis has to our knowledge not been done before, so we admit we were not sure what our expectations would be. As the clustering relationships of Figure 4 are the most biologically interpretable aspects of the work, so we have added some text to the Discussion: "Cancers from a common tissue, such as (KIRC, KIRP, KICH), (LUAD, LUAC), and pan-GI cancers are good predictors of each other, and there are also significant similarities within adenocarcinomas and carcinomas, respectively. These findings further demonstrate that cancer tissue of origin and glandular derivation are reflected in image-based cross-classification results. "

More broadly speaking, we agree that improved biological interpretability is a key next direction for not only our work but many CNN-based approaches. We expect that our cross-classification approach will enable new techniques to combine imaging and genetics data for such understanding, though that is beyond the scope of the current manuscript.

I have not further comments. Thanks for the chance to review this very interesting work.

Reviewer #2 (Remarks to the Author):

This manuscript describes an analysis of 27,815 images derived from 19 different cancer types from TCGA using a convoluted neural network (CNN) approach to distinguish cancer vs. normal, achieving an AUC=0.995, and discriminate specific subtypes (variously defined) with an AUC=0.87. CNNs trained on one tissue were effective in distinguishing cancer vs. normal in test sets derived from cancers not included in the training set. Highest levels of discrimination were achieved for breast, bladder and uterine cancers (versus normal). The CNN achieved discrimination for TP53 mutant cancers from wild type (AUC=0.65-0.80). The manuscript concludes that the universality of performance of the CNN across different tumor types suggests

a capacity to perform “cross-comparisons to reveal conserved spatial biology”. The manuscript attempts to address under-explored areas of potential interest: 1) how CNNs trained on one cancer type perform on other cancer types; 2) the efficacy of transfer learning and 3) factors that predict classification accuracy. The analysis also addresses how cellular versus inter-cellular tissue regions impact classification performance. This is an ambitious project with substantial potential; however, the breadth of the work comes at expense of depth, which limits interpretation of the data. At this stage in the development of CNN approaches, exhaustive interactions with “pathologists” (which can be non-pathologists with relevant expertise) is generally required to turn data into knowledge that can be leveraged to improve biological understanding or achieve clinical translation. This project includes world class senior pathologists, but would benefit from interactions with pathologists who have the time to work closely with the images and data. Perhaps most importantly, it is useful in such analyses to assess independent and meaningful gold standards, which are not related to classification of training and test sets. The specific

Analysis of TP53 status is a good example, but there are innumerable others that could have been considered, and this would assess whether the CNN provides information that microscopists cannot. The use of TCGA in this experiment is questionable. TCGA is a skewed collection of cancers that were identified to develop a working molecular analysis of all cancer types. Studies that leverage these strengths make best use of the resource (analysis of pan-omics data for hypothesis generating). The current analysis focuses mainly on comparing a CNN classifier to pathologists’ interpretations of morphology and representativeness of pathology is not the strength of TCGA.

We thank the reviewer for this valuable perspective. As the reviewer remarks, we indeed initiated this study to determine what can be learned from a “breadth” approach. We also agree that to achieve clinical translation, depth-oriented analysis is essential. We believe these approaches to be complementary, similarly to their dynamics in the fields of cancer genomics/genetics. A related point is that because this study considers such a broad collection of data, substantial effort is needed to identify aspects (e.g. TP53 status) that should be prioritized for detailed analysis. We therefore appreciate the reviewer’s identification of major points of concern. Our enumerated responses are provided below.

Major Points

1. The source of TCGA image sets and the features of these images are not adequately described. For example, TCGA was restricted to larger tumors that could be collected fresh and snap frozen for molecular analysis. TCGA samples were excluded if percent tumor composition did not meet specific thresholds. Further, image quality may have been issue, specifically for frozen sections; although such images are of generally reduced quality compared to sections of unfrozen tissue, nearly all frozen tissues are made from cancers, and the artifacts of freezing could trigger the correct classification of cancer by the CNN. There are likely many issues related to size, quality and preparation of images that could affect CNN performance, and impact either the accuracy of generalizability of the results. No analysis is perfect, but knowledge of these factors is helpful in putting the results into perspective.

We agree that such quality issues are important, and here we provide a more thorough description of the slides used. In our tumor/non-tumor classifier we have only used frozen slides. TCGA FFPE slides are predominantly tumor, and due to this we did not use FFPE slides for the tumor/normal classification task. Frozen slides have intermediate to large tumor sizes and exhibit a wide range of purity values (TCGA “average percentage of tumor cells”), as shown in Figure S2. Thus we did not filter the slides for only those with high purity, unlike some other recent studies. 24% of the frozen slides are normal (non-tumor) and the remaining are tumor. Regarding associations of performance versus auxiliary variables, our models were able to correctly distinguish tumor from normal slides with high AUCs in the reserved test data (0.995 ± 0.008). As such, there is negligible variability in tumor/normal predictive accuracy for any partitionings of the data into substantial subgroups. We have edited the Methods to clarify the characteristics of the slides (p.11): “24% of the frozen slides are labeled normal (non-tumor) and the remaining are tumors. For each tumor type, the annotated purity values (TCGA “average percentage of tumor cells”) span a broad range of values (Figure S2). “

We were able to compare FFPE and frozen samples for the subtype classification problem, as this uses only tumor samples and there are substantial numbers of both frozen and FFPE samples across subtypes. When we trained subtype prediction models, these models worked comparably in both frozen and FFPE samples. This indicates that the effectiveness of the CNN methods is a property of the subtype rather than the slide preparation method. Please see also the response to Reviewer 1, question 1 for details. We have added a new Figure S4 and also edited the text (p.5):

“The images used in the subtype analysis were from a mixture of frozen and FFPE samples. Although FFPE samples are preferred because they avoid distortions caused by freezing, we tested whether the CNNs were able to classify subtypes for each sample preparation (Figure S4). The CNNs classified the FFPE and frozen samples with comparable accuracy, with the same tumor types doing better (e.g. kidney) or worse (brain) in each. Correlations between classification AUCs were high across the two sample preparations ($r = 0.87$ for macro-averages; $r=0.78$ for micro-averages). As expected, FFPE-based classifications tended to be better, notably for brain and sarcoma samples.“

Our response to Reviewer 1, Question 3 may also be of interest. There Reviewer 1 asked us to discuss histopathological considerations that impact classification accuracy across tumor types.

2. What was the source of the images of “normal”? This is an incredibly important issue, with substantial implications for interpretation of the data. How were normal samples selected and what can we say about the composition of these images? For example, normal breast age-matched to breast cancer is largely fat and contains few epithelial cells, whereas cancer tissues are highly cellular. Thus, what appears to represent accurate discrimination of breast cancer from normal breast tissue could be driven partly (or largely), but a simple assessment of cell density, which is not a great achievement for image analysis with or without the application of a CNN. Similar arguments would apply at the level of a slide or an image tile within a slide. Using “normal” TCGA tissues is not ideal because these are normal tissues associated with cancers.

Data show that such tissues are often not normal, using various approaches, ranging from morphological to molecular.

We agree that the adjacent normal slides may be influenced by nearby cancer tissue. TCGA does not have true normal slides, and external sets would have high potential for batch effects. This is why we chose the TCGA “normal” set as an approximation of normal, as well as because of TCGA’s large data size and tissue matching. We have noted important caveats in the revised Methods (p.11):

Here the term “normal” is used to refer to the adjacent normal slides of a tumor. Although such regions are not tumor, some may be influenced by proximity to cancer tissue, and a more precise term might be “non-tumor”. Here we use the “normal” label for convenience and to be consistent with TCGA terminology.

Regarding the issue of fat, tiles with excess white space (which fat will tend to appear as) are filtered out before training the classifiers, which should limit their impact. Furthermore, the cross-classification results suggest that fatty tissue would not fully explain the efficacy of the tumor/normal classifiers. Different tissue types vary substantially in fat content, yet many cross-classifications are highly effective. For example the tumor normal classifier trained on breast cancer can also separate tumor and normal slides of bladder cancer despite their differing fat content. We have made two edits:

P. 12, Methods: “This step removes regions without tissue and also limits regions with excess fat.”

P.9, Discussion: “Cross-classification for tumor/normal status was successful across most tissues, despite the variations in native tissue morphology and fat content.”

3. Without knowledge of the composition of various test sets, the interpretation of AUC is unclear. While true that the sensitivity and specificity of test performance are independent of the case vs. control percentages of a sample, what is more important is the predictive value of a classifier. For example, even a high specificity would give a poor positive predictive value if the prevalence of the disease is low.

A. These considerations could also impact the assertion that certain cancer types are “canonical” examples of the power of CNNs to offer important classifications. We do not know the composition of cancer vs. normal per tumor type or have a sense of the composition of tumors and normal samples. In the uterus, the definition of “normal” is as troubling as for breast; normal could mean atrophic, proliferative, secretory or something else and each of these morphological appearances is quite distinct with regard, cellularity, cell organization, cytology, etc.

The reviewer is correct that high specificity and sensitivity are both important in evaluating the performance of a classifier. This is why we have reported areas under the curve (AUC) for sensitivity vs specificity plots. AUC can reach a value close to 1 only if the classifier can simultaneously achieve both high sensitivity and high specificity. In Figure 2d we have reported

two types of 'per-slide' AUC: the ROC AUC (derived from sensitivity/specificity) and PR AUC (derived from precision/recall). Both of these measures are greater than 0.97 in all the considered tumor types, showing that the classifiers can simultaneously achieve high sensitivity+specificity or high precision+recall. The within-slide behavior is more challenging, and this is quantified in Fig 2b for tumor/normal classification. Still, there we show that even 'per tile' precision, recall, specificity, and accuracy are all greater than 0.7 (at our default threshold for calling a tile to be tumor) in all the tumor types considered.

Regarding the evaluation of certain cancer types as 'canonical', we agree that this assessment is dependent on the datasets used for the analysis. TCGA is among the largest, most systematically curated datasets. However, it focuses on common cancers, and there are cancer types and "normal" tissue classes not explored within it. We have added a comment about this limitation to the Discussion section. Note also that we have performed a new analysis on CPTAC image data in response to Question 5a.

P.9: "While such results are subject to the limitations of the data used, TCGA is one of the largest available sets, as we were able to validate the effectiveness of TCGA-trained classifiers on independent CPTAC lung cancer sets as well."

4. The section of tumor-predicted fraction and tumor purity requires explicit definitions of each term. What is the gold standard and why should we accept that the propose definition is the most accurate metric? This is a complicated area, and optimal metrics may vary by applications. If the experiment is DNA based, then knowing percent DNA purity is critical (cancer vs. not cancer), whereas if the issue is related to percentage of cells demonstrating a particular marker (e.g. immunostain), then percent of cells that is tumor is critical.

TPF is the fraction of non-background tiles in an image that are predicted by the CNN to be tumor. We have edited the Methods section (p.12: CNN architecture and training) to make this clearer. For tumor purity, we use the TCGA-annotated quantity "avg_percent_tumor_cells," which is based on histopathological analysis, which we have clarified in the Methods (p.11: Sample selection for tumor/normal classification). See also the response to Question 3 from Reviewer 1.

5. The conclusion that cancer vs. non-cancer is a fundamental distinction that can be achieved by training a CNN is perhaps intuitive. These "classes" are vastly different with regard to nearly any feature: morphological, molecular, cellularity, etc. Demonstrating the universality of cancer features across tumor subtypes is of some interest, but it would be useful to provide greater insights into the drivers of this distinction if possible. What are the features of the cancers misclassified? Were they reviewed by a pathologist?

We agree with this concern. We observed that most misclassification was for adjacent normal slides with unexpectedly large predictions for TPF. Our pathology team assessed such slides, and we observed they often suffer from poor quality, tissue folding, or excessive tissue damage related to freezing. Below is an example of a TCGA adjacent normal in which the CNN predicts

many regions to be tumor (indicated in red in the heatmap). The H&E shows that the incorrectly predicted regions appear to suffer from freezing damage. Thus we believe our misclassifications stem from cases where there are sample preparation artifacts, rather than the artifacts being responsible for the correct classifications.

Figure S3 has been added, showing this example as well as others. The following text has been added to the manuscript (p.4):

“Most misclassification was for adjacent normal slides with unexpectedly large predictions for TPF. Manual pathology review indicated that such slides often suffer from poor quality, tissue folding, or excessive tissue damage related to freezing (see Figure S3 for examples).”

5a. The TCGA set is peculiar and not ideal for making generalizable conclusions about many issues in pathology and biology. However, it may be useful for hypothesis generating. Overfitting is a major problem in developing CNNs, and the peculiarity of TCGA samples weighs on this issue. It would be much stronger to test the CNNs on non-TCGA samples to accurately assess performance. Even if this were done in a limited way, it would help understand these data.

We agree with the reviewer that an external validation on non-TCGA samples would be valuable. We have applied the TCGA trained classifiers to LUAD and LUSC cancers of the Clinical Proteomic Tumor Analysis Consortium (CPTAC) dataset (note: CPTAC denotes lung squamous cell carcinoma as “LSCC”, while TCGA uses “LUSC”). We found that the TCGA-trained LUAD and LUSC classifiers have very high validation AUCs on the CPTAC images. We also observed high correlations between in the cross-classification AUCs when applied to the TCGA test and CPTAC validation sets. To describe this, we have added a new section (p.7. See revised manuscript file for the new figures: Figure 6 and Figure S7):

“Validation of cross-classification relationships using CPTAC images

To validate the trained CNNs and their cross-classification accuracies we applied them to the LUAD and LUSC slides of the Clinical Proteomic Tumor Analysis Consortium (CPTAC) dataset (see Methods). TCGA-trained LUAD and LUSC classifiers were highly effective on the CPTAC LUAD and LUSC datasets (Figures 6a,b). The TCGA-trained LUAD and LUSC classifiers have

validation AUCs of 0.97 and 0.95, respectively, on the CPTAC-LUAD dataset, and have validation AUCs of 0.97 and 0.98, respectively, on the CPTAC-LUSC dataset. Both of the TCGA-trained CNNs yielded well-separated distributions of TPF between CPTAC tumor and normal slides (Figure S7). CNNs trained on other TCGA tissue types were also relatively effective on the CPTAC sets, with average AUC of 0.75 and 0.73 when applied to the CPTAC LUAD and LUSC image sets, respectively. This was lower than the performance of the TCGA-trained classifiers on the TCGA LUAD and LUSC sets (average AUC 0.85 and 0.90, respectively), suggesting that cross-classification is more sensitive to batch protocol variations. However, the correlation between AUCs on the TCGA and CPTAC sets was high (Figure 6c,6d: LUAD: $r=0.90$, LUSC: $r=0.83$), indicating that relationships between tumor types have a clear signal despite such sensitivities.”

6. Interpretation of data on a per tile basis may be limited by lack of description of the size of tissues images, magnifications, and purity, etc.

We agree with this point. Tile-scale biological interpretation is considerably more complex than can be resolved in this paper, which mainly deals with tumor/normal evaluations and some special cases. Extensive tile-level pathologist annotations are at the minimum necessary for this, and these are not provided by TCGA. We have clarified this limitation in the Discussion.

p.11: “Finally, to improve tile-level understanding using these approaches, further fine-grained pathological annotations with concomitant hypothesis development from the community are vital, e.g. through extended curation of TCGA and other sets.”

7. The ability of the CNN to discriminate cancer with TP53 mutations is of interest, but the analysis does not provide depth of understanding. How does this relate to grade or tumor size or cellularity? Has anyone visually compared these tumors in relation to CNN classification? Are there penultimate layers of the CNN that can provide analytical data that could offer insights (morphometric features)?

To assess the importance of pathological features, we trained a random forest model for classification within and across the 5 tumor types used for the TP53 analysis. We did not use tumor grade since it had missing values for most samples. The random forest was made up of an ensemble of decision trees having various thresholds for two features: *purity* and *stage*. The random forest model yielded lower classification performance (See Figure S10) compared to the CNN model, indicating that the CNN model uses information more sophisticated than tumor purity and stage. We have added a note about this on p. 7: “The CNNs achieved a higher AUC compared with a random forest using tumor purity and stage for TP53 mutation prediction (see Figure S10), suggesting the CNNs use more sophisticated morphological features in their predictions.”

Our pathology team also reviewed slides in which TP53 status had been classified, such as those in Figure 8. However, there are no known visual features for TP53-mutant cells in H&E data. Given that fact, we believe that the CNN is not classifying TP53 status at the individual cell level, but rather that the CNNs are making predictions based on tumor microenvironmental features correlated with TP53 mutant status. The exact nature of such features remains unclear,

though we agree this is an important problem for future studies. We have added a comment in that section of the text:

p.7: “A caveat to these analyses, however, is that the spatial variation within heatmaps may reflect TP53mut-associated microenvironmental features rather than genetic variation among cancer cells.”

As the reviewer suggests, the penultimate layers contain information (though this may depend on the CNN architecture), and development of approaches to interpret them requires significant research effort. We are currently pursuing this through attention-based analysis, associations between penultimate layer features and genetic features, and other methods.

Minor Points

1. Philosophical question: Is the goal of the current project to demonstrate how CNNs can reduce human effort, as implied by the introductory statement, “Manual inspection is time consuming ...” or to demonstrate the potential for computers to extend the horizons of morphological characterization of tissues or both?

Our goal for this project has been to “extend horizons” by showing new data analysis approaches involving training and comparison of diverse CNNs, though the aspects the reviewer raises are entangled. The ability to extend horizons comes from the increasingly large data collections that can be rapidly compared among one another using CNNs. These collections are so large that it would be infeasible to do such comparisons manually. Thus, the question of whether human effort is being reduced is confounded by the fact that CNN comparisons are not straightforwardly convertible into manual tasks.

2. The Abstract refers to “slides” but the analysis was performed on scanned images, a non-trivial distinction.

We thank the reviewer for catching this. We have corrected this in the text.

3. The Abstract refers to “accuracy” but the gold standard of accuracy must be stated.

We have adjusted the abstract to clarify that all “gold standards” are pathologist annotations. Details are in the Methods.

4. The Abstract conclusion that CNNs may perform “cross-comparisons to reveal conserved spatial biology” is rather vague or obscure, without specific meaning. A more tangible, if somewhat unglamorous sounding statement, would be more meaningful. CNNs can distinguish cancer from non-cancer based on their potential to appraise features used in histopathologic diagnosis, such as cellular density and organization (if this is what is meant).

We recognize the importance of the reviewer’s critique. However, although we can quantify the empirical accuracy of CNN predictions, a limitation of current CNN methods is that they do not output the biological features responsible. Further research approaches are necessary to

confirm such features. We have adjusted the statement in consideration of the reviewer's comment to: "These results demonstrate the power of CNNs not only for histopathological classification, but also for cross-comparisons to reveal conserved spatial behaviors across tumors."

5. The section "Neural network classification of cancer subtypes" is confusing.

- A. Some cancers list subtypes that are chosen for discrimination, such as lung adenocarcinoma vs. squamous cell carcinoma, but others are not (e.g. sarcoma). The text and figures would benefit from clarification and revision to increase consistency and clarity.
- B. Lists of cancer subtypes included in analyses might be relegated to supplement.
- C. Reference to cervix and adenoma, presumably should read adenocarcinoma.

A. We included the following in the revised text:
"Since SARC does not fit either category, and ESCA contains a mixture of both categories, these two cancers were labeled as 'other'"

- B. We have moved the names to the corresponding section in the Materials and Methods
- C. The reviewer is absolutely correct. We have fixed this in the text now.

6. The manuscript includes several statements in which performance of the CNN as a classifier is invoked as a means of providing biological insights, but such statements are not specified. What sorts of biological insights are provided and do these expand our knowledge of cancer biology?

As discussed above, CNN approaches yield findings about predictive accuracy, making the insights fundamentally quantitative. For example, our results on how predictors trained on one cancer type work on others provide evidence for histopathological similarities that would be non-obvious otherwise (Figure 4). We have made additional edits to the Discussion to better summarize the main results:

P.9: "Cross-classification for tumor/normal status was successful across most tissues, despite the variations in native tissue morphology and fat content. These studies showed that tumor images have robust intra-slide structure that can be consistently identified across CNN classifiers."

P.9: "Cancers from a common tissue, such as (KIRC, KIRP, KICH), (LUAD, LUAC), and pan-GI cancers are good predictors of each other, and there are also significant similarities within adenocarcinomas and carcinomas, respectively. These findings further demonstrate that cancer tissue of origin and glandular derivation are reflected in image-based cross-classification results."

7. Use of the word "cohort" is unfortunate. TCGA is not so much a cohort of any sort, but rather a tissue collection with extremely modest annotation of patients and clinical factors. Indeed this is a major limitation of TCGA.

We have adjusted the language to change “cohort” to “cancer type,” “cancer,” or “set” where appropriate.

8. A better description of “transfer learning” would increase understanding for readers less familiar with CNN as applied to morphology.

We have expanded the section on “Transfer learning as a common feature extractor” in the Discussion section (p.10). We have also added a citation to a recent paper that describes transfer learning in the medical imaging field.

Raghu, M., Zhang, C., Kleinberg, J. & Bengio, S. Transfusion: Understanding Transfer Learning for Medical Imaging. arXiv.org cs.CV, (2019). <http://arxiv.org/abs/1902.07208v3>

9. Table 1 reference to slides in relation to TP53 analysis is not necessarily apt. TCGA data imply that somewhere in the tumor (the part that was ground up and tested) a mutation was or was not found. Unfortunately, we do not know for sure if there was a mutation in the part of the tumor that was used to create the image of the morphology do we?

TP53 is commonly an early mutation in tumors, in which case it should be shared across all regions of the tumor. Still, as the reviewer suggests, it may be spatially heterogeneous in some cases. Therefore we expected that the method would work better when the minor allele frequency (MAF) of TP53 was higher, as in such cases the TP53 mutations would be more likely to be ubiquitous throughout the tumor. To test this, we conducted a new analysis using a minimum threshold for mutation frequency (For an ubiquitous mutation, TP53 would have MAF=0.5 in a heterozygous diploid. However, variations in sample purity and copy number aberrations can modulate this value). We tested our CNNs within each cancer type, requiring a stringent minor allele frequency (MAF > 0.25). Our model performs better on these higher mutation frequency cases in all cancer types considered (See new Figure S11), consistent with the hypothesis that TP53 is easier to detect when it is more spatially ubiquitous. We also analyzed whether restricting to IMPACT=HIGH cases would improve AUC values. However, different cancer types showed varying effects, indicating that our original constraint of IMPACT=MEDIUM or HIGH was sufficient for the study.

We have added a sentence to the TP53 section of the main text (p.7: “We also observed that CNNs were able to more accurately identify tumors with TP53 mutations when the allele frequency of the mutation was higher, suggesting that prediction is easier when the tumor is more homogeneous (Figure S11).”) to address these points.

Reviewer #3 (Remarks to the Author):

The authors evaluate the role of deep learning using digital images of tumors from different TCGA cohorts and indicate that a convoluted neural network (CNN) training in one tumor type can recognize and accurately classify tumors from normal tissue and TP53 mutation status in a different organ-type cohort. The authors conclude that this cross classification has biological relevance.

1. The article is difficult to understand and read as there are many abbreviations for tumor types used that are not clearly indicated in the text.

We apologize for the confusion. We have edited the Methods section to provide definitions of tumor types and subtypes. The tumor type abbreviations follow TCGA terminology, available at: <https://gdc.cancer.gov/resources-tcga-users/tcga-code-tables/tcga-study-abbreviations>
We have edited the Methods to provide this link.

2. The authors use many different tumor types including sarcomas and brain tumors, but concentrate their efforts throughout the article on epithelial tumors (carcinomas). It would be better to report only work on carcinomas, as the non-carcinoma types do not perform well in their model.

While tumor types have different results for different analyses, there are concrete results for non-carcinomas as well. For example, the tumor/normal self-classification is extremely accurate for every tumor type considered (Figure 2d, all AUCs are >0.97). Consequently, we believe it is informative for readers to see the non-carcinomas compared with the other tumors, particularly given that comparative analysis is the motivation for the paper.

3. Fig 2: No all tumor types indicated are described in the text. The text indicates that testicular tumors were included, but they are not represented in the figure. Testis is indicated in Fig 3, but other tumor types are missing from that figure.

As noted above, we have now provided a link to the definitions of all TCGA tumor types shown in Figure 2 in the Methods (p.11). Figure 3 focuses on the tumor types that can be broken down into more specific subtypes. So tumor types that do not have subtype annotations are not shown in Figure 3. Also, the reviewer is correct that the tumor types in Figure 3 are not a pure subset of Figure 2 (as the reviewer notes for testis). This is because the Figure 2 analysis requires both tumor and normal tissue, while the Figure 3 analysis requires only tumor tissue. So the sample criteria for each analysis are different. We have edited the Methods section to make this logic clearer.

P.12: "Note that the subtype analysis requires only tumor tissue, so it includes some cancers that were not included in the tumor/normal analysis because of minimum data requirements on the normal samples."

4. Fig 4: similar issues with abbreviations and tumor type. In addition, the authors suggest that clustering of lung and kidney; and breast and GYN tumors. However, there are no further explanations. Was the clustering related to specific tumor types, eg. endometrioid adenocarcinoma of the uterus versus colon cancer? Lung cancer and kidney cancer do not seem to share morphological similarities. Could nuclear hyperchromasia be responsible for the clustering?

We have addressed the issues regarding abbreviations and tumor types above. With regard to the clustering, unfortunately the annotations of endometrioid adenocarcinoma of the uterus that the reviewer describes were not available in TCGA and so were not able to analyze this point. However, to consider the review's concerns our team has performed additional manual review on a subset of samples that were misclassified by our tumor/normal CNNs (see Figure S3. Responses to Reviewer 1, Question 3 and Reviewer 2, Question 5 may also be of interest.) We observed that misclassified slides were often related to slide quality issues such as tissue folding or freezing artifacts. In some cases we did see nuclear hyperchromasia in misclassified regions as well, though these were anecdotal.

More broadly, we agree that delineation of sample subsets and/or features responsible for the clustering is important, but that will require considerably more extensive manual pathology analysis as well as exploration of new classifiers trained on combinations of datasets. These are of interest but are beyond the scope of this work. The revised Discussion includes some new text related to these points:

P.9: "Overall, the high levels of cross-classifiability suggest that it will be possible to combine images from multiple cancer types to extend and refine training sets. Investigations into the optimal combinations of sets (both positive and negative) may be useful for improving a variety of classification tasks."

P.11: "Finally, to improve tile-level understanding using these approaches, further fine-grained pathological annotations with concomitant hypothesis development from the community are vital, e.g. through extended curation of TCGA and other sets. "

5. Can these findings be validated in a non-TCGA cohort?

This question was also asked by Reviewer 2, Question 5a. We agree with the reviewer that an external validation on non-TCGA samples would be valuable. We have applied the TCGA trained classifiers to LUAD and LUSC cancers of the Clinical Proteomic Tumor Analysis Consortium (CPTAC) dataset (note: CPTAC denotes lung squamous cell carcinoma as "LSCC", while TCGA uses "LUSC"). We found that the TCGA-trained LUAD and LUSC classifiers have very high validation AUCs on the CPTAC images. We also observed high correlations between in the cross-classification AUCs when applied to the TCGA test and CPTAC validation sets. To describe this, we have added a new section (p.6. See revised manuscript file for the new figures: Figure 6 and Figure S7):

"Validation of cross-classification relationships using CPTAC images

To validate the trained CNNs and their cross-classification accuracies we applied them to the LUAD and LUSC slides of the Clinical Proteomic Tumor Analysis Consortium (CPTAC) dataset (see Methods). TCGA-trained LUAD and LUSC classifiers were highly effective on the CPTAC LUAD and LUSC datasets (Figures 6a,b). The TCGA-trained LUAD and LUSC classifiers have validation AUCs of 0.97 and 0.95, respectively, on the CPTAC-LUAD dataset, and have validation AUCs of 0.97 and 0.98, respectively, on the CPTAC-LUSC dataset. Both of the TCGA-trained CNNs yielded well-separated distributions of TPF between CPTAC tumor and normal slides (Figure S7). CNNs trained on other TCGA tissue types were also relatively effective on the CPTAC sets, with average AUC of 0.75 and 0.73 when applied to the CPTAC LUAD and LUSC image sets, respectively. This was lower than the performance of the TCGA-trained classifiers on the TCGA LUAD and LUSC sets (average AUC 0.85 and 0.90,

respectively), suggesting that cross-classification is more sensitive to batch protocol variations. However, the correlation between AUCs on the TCGA and CPTAC sets was high (Figure 6c,6d: LUAD: $r=0.90$, LUSC: $r=0.83$), indicating that relationships between tumor types, have a clear signal despite such sensitivities."

Reviewer #4 (Remarks to the Author):

The authors propose a deep learning network for binary and multi-class classifiers for 19 different cancer types. A large number of experiments on a large number of multi-tissue TCGA dataset are reported. Using the standard Inception-v3 network for both binary and multi-class classification of TCGA cases into tumor and normal and subtyping of tumors, the authors report AUC scores of 0.995 ± 0.008 and 0.87 ± 0.1 for binary and multi-class classification, respectively. They further reported results of cross-tissue type performance analysis of their classifiers again showing results in AUC score of 0.88 ± 0.11 . They conclude that some cancer types are relatively easy to classify as compared to others; TP53 mutation can be predicted from histology images with AUC score range 0.65-0.80.

I have following concerns regarding the manuscript:

(1) In Binary classification, all tiles of the normal case are considered normal and all tiles of the malignant case are considered as tumors. Given a tumor sample may contain a large number of normal tiles and only a handful of tumor tiles, how can that approach be justified?

The reviewer is correct that in binary classification all tiles of the adjacent normal slide are labeled as normal, and all tiles of the malignant cases are labeled as tumors. It is also correct that several tiles of a tumor slide may not contain any tumor related morphological features, meaning they belong to the adjacent normal class. As TCGA tumor slides are rather pure, i.e., tumor cellularity is high, we expect such tiles to comprise only a small portion of training data. Similarly, adjacent normal slides may infrequently contain small tumor regions. Therefore, training data in each class is slightly polluted by tiles that truly belong to the other class. However, as long as this pollution is not high, features will correctly be associated with class labels. The theoretical analysis of the problem and why reliable classification is possible is studied in detail in the reference below. We have added a comment about this to the Discussion (p.10: 2nd paragraph of "Transfer learning as a common feature extractor"). A common technique to handle this problem is multi-instance learning, which is discussed in the same section of the manuscript.

Cannings, Timothy I., Yingying Fan, and Richard J. Samworth. "Classification with imperfect training labels." *Biometrika* 107.2 (2020): 311-330.

(2) It is stated that the tumor-predicted fraction (TPF) in each slide is then "used as a metric". It is stated that per-tile classification ROCs were calculated based on thresholding softmax probabilities and per-slide classification ROCs were based on voting on maximum softmax

probability. Again, a slide can be labelled as a tumor slide even though a very small number of tiles (even only one tile) is labeled as tumor. How would that work?

We apologize that our prior wording about “voting on maximum softmax probability” was confusing. We agree with the reviewer that the original manuscript suggested TPF is the maximum probability of being tumor across all tiles, but that was a mistake in our explanation. The procedure is that each tile is associated with the class having the maximum softmax probability (i.e. tile values are binarized). Then the voting scheme is based on the ratio of tiles labeled as tumor. This ratio defines TPF. To rectify this issue the Methods section has been revised accordingly:

p.12: “To compute per-slide classification ROCs, each tile is associated with the class having the highest softmax probability. Then the fraction of tiles labeled as tumor, i.e., TPF (tumor-predicted fraction), is used to distinguish classes at the slide level.”

(3) A standard Inception v3 network trained on ImageNet is used with a basic transfer learning method that trains the final couple of layers on the histology image data for both binary and multi-class classification. Although, high AUC score values support the claim that transfer learning work, a discussion of how a network trained on natural images work on histology images, when the contents of two types of images are quite different, is missing.

We agree that a discussion on the ability of transfer learning models pre-trained on natural images to detect tumors would be valuable. The paragraph below has been added to expand the Discussion section on “Transfer learning as a common feature extractor”:

P.10: “That being said, the ability of transfer-learning based models to classify tumors remains noteworthy. Even though the Inception network never used pathology images in pre-training, the large set of image-net pre-training images across diverse object classes still led to pre-trained feature representations encoding information salient across cancer types. Further incorporation of histopathological sets during pre-training may improve the resolution of classes with more subtle differences, such as those that differ by single mutations, and this will be an important topic for future study. Continued development of transfer learning methods for biomedical image analysis (Raghu et al. 2019) and investigations into the general ability of effective representations to encode information for various tasks, as has been discussed in detail by Bengio et al. (2013), will both be pertinent.”

A more detailed discussion on efficient feature construction is beyond the scope of the current manuscript, and we refer the reviewer to the cited reference for further discussion: Bengio, Yoshua, Aaron Courville, and Pascal Vincent. "Representation learning: A review and new perspectives." *IEEE transactions on pattern analysis and machine intelligence* 35.8 (2013): 1798-1828.

(4) For the tumor/normal classification CNN, all training data is based on frozen slides and results in AUC of 0.995, whereas in cancer subtype classification both FFPE and Flash Frozen are used, the AUC is decreased to 0.87. A discussion of the drop in performance is missing.

We thank the reviewer for this comment. We have conducted new analyses which indicate that subtype classification is comparably accurate on each of FFPE and flash frozen samples. See supplementary Figure S4. A full description of this is provided in the response to Reviewer 1, Question 1. This suggests that the lower AUCs for subtype classification are because subtype classification is inherently more challenging than tumor/normal classification, rather than due to frozen vs. FFPE concerns. The following paragraph has been added to the paper:

pp.4-5: "In contrast to tumor/normal classification achieving high AUC's across all cancer types, subtype classification AUCs are lower and span a wider range. This suggests that subtype classification is inherently more challenging than tumor/normal classification, with a narrower range of image phenotypes."

See also the subsequent paragraph in the text, which is related to the response to Reviewer 1, Question 1.

(5) For TP53 mutation, AUC scores below 0.75 could not be considered reasonable. What is the F1 score like? If that's below 0.7, a histology based test would be of no use.

We have added a supplemental figure that reports cross- and self- classification F1-score values (See Figure S12) from balanced deep learning models (with 95% CIs). Self-classification values range from 0.63-0.75. We agree that the classification accuracy is not at a level for practical use. Our purpose for the TP53 analysis was to show that cross-tumor classifications have considerable predictive information even for individual driver mutations, a basic research result which has not been previously known. To develop a histology-based test, significant advances would be needed with the current results suggesting approaches to do so, e.g by testing different neural network architectures, hyperparameters, and combinations of training data across histologies. We have updated the Results to explain this (p.7): "These AUC values are not sufficient for practical use, though the positive cross-classification results suggest that it might be possible to combine datasets to increase accuracy (see Discussion)."

(6) Deep learning-based approaches usually require a validation set along with training set for hyperparameter tuning and optimal model selection. Authors have used test set as validation set for "Tile-based purity estimation". However, it is not clear that how they selected the optimal models for all other experiments.

Validation sets are typically used to 1) provide an unbiased evaluation of model fit during the training and 2) to tune model hyperparameters. In contrast, the test data provides an evaluation of model fit after training and is the main measure of model performance. For fine-tuning hyperparameters and model selection (e.g., regularization parameters) it is particularly important to use a validation set.

However, none of our models have any hyper parameters to tune and we are not selecting against competing models. Given the limited number of available samples for training in many of the cancer types, and given the robust model performance on test data, we opted not to use a validation set and instead reserve the data to train the models. We would like to note that multiple train/test split runs (3 total per cancer type) and the newly added independent test data from CPTAC demonstrate that the models are not overfitted to the training data and are generalizable.

The CPTAC image data are an independent new validation set for which we have shown that TCGA-trained classifiers successfully predict tumor/normal status. For full details, see the response to Reviewer 2, Question 5a.

(7) Authors listed “transfer learning as common feature extractor” as one of their findings which is already well studied topic and number of methods have used transfer learning based features in multiple image modalities.

We recognize that we are not the first group to use transfer learning, and our prior submission referenced several prior papers about this topic (e.g. Coudray et al. 2018, Xu et al 2019, Zhu et al 2011). For clarity, we have edited the introduction to emphasize the prior use of transfer learning by Coudray et al, as our work builds on their architecture. The Discussion section on “Transfer learning as a common feature extractor” is not a finding, but a summary of issues related to transfer learning. We have edited the wording of this section to make this clearer, including addition of more references.

(8) Since frozen slides alter the tissue appearance drastically, therefore, in practice, these are not considered suitable for computational analysis. Therefore, a separate experiment with only FFPE based training and testing of the pipelines is highly recommended.

We agree with the reviewer that frozen slides suffer tissue damage. For tumor/normal classification, unfortunately, TCGA does not contain sufficient normal FFPE slides for this, so we were unable to compare FFPE and frozen for that analysis. However, for subtype classification, we have added new analyses showing that subtype classifications are comparable in both FFPE and frozen tissues. This is discussed in detail in the response to Reviewer 1, Question 1 so we refer the reviewer to that description. These results show that FFPE and frozen images retain enough similarities for the CNNs to have cross-platform utility.

(9) Similarly, the cross-classification between tumor types is recommended to be evaluated with FFPE samples only. This is important to establish that the flash-frozen special appearance and resulting tissue damage is not the dominant factor in discriminating normal vs tumor slides and so on.

As described in point (8), unfortunately this is not possible as TCGA does not contain sufficient adjacent normal FFPE slides. However, our pathology team has further manually reviewed

slides that are misclassified by the CNNs. We have observed that freezing distortions tend to cause misclassification, rather than being responsible for the correct classification. See response to Reviewer 2, Question 5 and also new supplementary figure S3.

(10) It seems that for every cohort, some of the images are used for training and the remaining for testing. A true cross-classification experiment would be leave-one-TCGA-cohort-out and reporting average results on those experiments. It is recommended to run cross-validation experiments for each classification task.

The reviewer makes an interesting suggestion that would be appropriate for performing cross-validation of a universal tumor-normal classifier, though that is different from the goal of the current work (we note we can already perform accurate slide classifications by using CNNs trained on the individual tumor types, so our understanding is that the reviewer is interested in how tumor types can be combined). The clustering structure of the cross-classification diagram (Figure 4) demonstrates that there are biological relationships between tumor types. Notably LIHC is quite different from other tumor types, and there are statistical associations within other biological groups (Pan-GI, Pan-GYN, adenocarcinomas, carcinomas, etc.). These results are already more specific than what would be revealed by a universal analysis. Interpretation of the suggested leave-one-cohort-out analyses would also be dependent on the inter-tumor relationships. So we agree with the reviewer that the combining of tumor types together is a potent idea, but the underlying challenge is that more specific hypothesis development is needed. We have edited the Discussion to add this important concept, (p.9):

“Overall, the high levels of cross-classifiability suggest that it will be possible to combine images from multiple cancer types to extend and refine training sets. Investigations into the optimal combinations of sets (both positive and negative) may be useful for improving a variety of classification tasks.”

(11) TP53 prediction is slightly more accurate using self-cohort than cross-classification, which is claimed to be the main contribution of the paper.

As discussed in point (5) above, the purpose of this manuscript has been to show that there is significant cross-tissue classification information from image data, which has not been previously demonstrated. Our work shows that data from both the original tissue type and other tissue types will be valuable for optimizing classifiers. Whether self- or cross- classification is more accurate has not been a point we considered unusually important. We are unsure which section of the paper conveyed that impression, so we have edited the Discussion section in consideration of the reviewer’s comment:

P.10: “Self- and cross-comparisons of classifiers can highlight robust spatial structures within tumor images (e.g. Figure 8), but interpretation remains a major challenge.”

Minor comments:

m1. Many cases in the TCGA cohort contains multiple slides, therefore, all slides of a case should be part of only one dataset/data-split either train or test. However, it is not clear from the text that whether authors have followed this protocol or not.

The original manuscript split the data at a slide level rather than at a patient (case) level. To address the reviewer's concern, we have re-trained and tested the classifiers on patient level splits of data. The AUCs are compared in Figure S1, showing that splitting the data at the patient level instead made little difference to classification AUCs. We have edited the text by adding the following to the Methods:

P.14: "**Patient level stratification:** We considered two ways of splitting data by patient for the analysis of Figure S1. 1. First, we considered two patient groups-- those with adjacent normals and those without. For each cancer type, 70% of patients in each group were randomly assigned for training, and the remaining 30% were used for testing. Slides corresponding to each patient, whether in train or test, were placed in their associated class, i.e. normal or tumor. This data split was denoted by the "patient level" stratification. 2. Alternatively, we restricted analysis to patients who only have adjacent normal and used the 70/30 split of patients. This split was denoted by "matched patient level" stratification."

Figure S1b has also been added to compare the slide level and patient-level splits. The figure caption describes the result: "Per-slide AUC values for tumor/normal classifiers for the slide level, patient level, and matched patient level splits of data. The difference between the patient level split and slide level split across all cancer types is -0.007 ± 0.02 , and the difference between the matched patient level split and slide level split across all cancer types is -0.002 ± 0.009 ."

The main text has been edited as well (p.4): "These results were for slide level test/train splits of the data, but splitting at the patient level instead had little effect on classification accuracy (See Methods and Figure S1b)."

m2. Figure 1a: it is quite difficult to infer the number of slides due to log-based scale. Therefore, bar-chart based visualization would be better.

As suggested, we replaced this with a linear scale plot in the new edition (the panel in question is actually Figure 2a).

m3. The step of TFRecord conversion (Figure 1a and 1b) in the deep learning pipeline is not well-understood, what is it and why it is introduced in the pipeline is not well explained, is it just an efficient implementation or if it has a role in the network prediction too? Though, it seems core TensorFlow data structure/type.

The purpose of creating tfrecords is to make an efficient implementation, as suggested by the reviewer. The illustration is designed to help the reader understand the pipeline. The following

paragraph was added to the Materials and Methods section, under step 7 of “pre-processing and transfer learning steps” (p.12):

“The TFRecord file format is a simple format for storing a sequence of binary records. TFRecord is TensorFlow’s native storage format and enables high data throughput which results in a more efficient model training pipeline.”

Reviewer #5 (Remarks to the Author):

1. While the study of distinguishing cancer tissues from normal is an interesting problem to test the methodology, it is of low impact, as one can imagine that this would be trivial for pathologists to do. Similarly, several other modalities (e.g. transcriptomics, epigenomics etc.), can be easily used to classify cancer vs. normal. Similarly, “classical” computer vision approaches using morphometric features etc. have been used on WSI image data, to classify cancer vs. normal with the same performance. The same holds for distinguishing cancer subtypes. So how surprising is it that CNNs can do this?

We agree it is not surprising that a CNN -- given a training set of say breast cancer and breast normal WSIs-- can be trained to successfully classify other breast WSIs; indeed, that is not the major point of this work. However we do believe it is a novel finding that one can train a CNN on breast samples, and then have it be successful on lung samples. The success of such cross-classification is the main contribution considered in this study.

Regarding prediction based on morphometric features, that approach has some advantages and disadvantages relative to deep learning, which we will not fully review here. Prior literature is already cited in the Introduction. Relevant to this manuscript, we note that hand-crafting of morphometric features is an expert activity for which it is challenging to avoid bias to specific tumor types. An advantage of the deep learning approach is that feature pre-specification is not necessary, and the same deep learning architecture can be used equivalently across tumor types. This makes it easier to perform comparisons without concern about whether a crafted feature can be measured in a consistent fashion across all tissue types.

2. It is not clear if the FFPE & frozen slides can be used together? They have very different potential for artefacts, and the results can easily be biased towards different way of storing and preparing the slide. Is there any data to show that models built on both do not suffer from these potential issues?

This question is related to reviewer 1:question1 and was also asked by other reviewers. We summarize here and refer the reviewer to that response for elaboration. The TCGA database has very few normal FFPE whole slide images. So for the tumor vs normal classification task we only used frozen samples, and it was not possible to compare FFPE to frozen for that task. However tumor samples were a mixture of frozen and FFPE, and we were able to compare FFPE to frozen for the subtype classification task. We have added Supplementary Figure S4, which depicts the performance of the model when the trained model (based on both FFPE and

frozen samples) is applied to just FFPE samples or just frozen samples. We find that FFPE and frozen samples have comparable classification performances.

3. Even though there is a statistically significant correlation between TPF and pathologist reported purity, actual correlation is still quite low, most cancers are in the 0.2 vs. 0.4 range. What can be concluded from this analysis? What happens if you correlate with genomic measures of tumor purity?

We agree that the moderate correlations between TPF and purity suggest that additional factors impact TPF. Please refer to the response to reviewer 1 comment 3, which describes potential confounding variables. In brief, we believe that the network may be incorporating cancer-related image features such as necrosis and lymphocyte infiltration into the prediction of tumor tiles, but the association between such variables and purity varies as a function of cancer type. Different pathology measures of purity are better associated with TPF in certain cancer types, suggesting approaches for refining the understanding of predictive features. Understanding of the biological features that determine TPF is a large project which we are now pursuing in additional studies (e.g. using attention-based methods, genetics/CNN associations, and non-linear correlation models, manuscript in preparation). Variations across pathologist assessments may also impact correlations.

Regarding genomic measures, one problem is that the sample used for genomic analysis may not be the same sample used for imaging. This is an important caveat to the comparisons with genomic assessments. Nevertheless, we considered two methods for computing genomic measures of purity: ABSOLUTE (Carter et al. 2012) and InfiniumPurify (Zhang et al. 2015). ABSOLUTE is a widely used method based on copy number data while InfiniumPurify uses methylation values. To simplify the analysis, we analyzed a single cancer type, BRCA. The correlations of TCGA-annotated purity vs. the ABSOLUTE and InfiniumPurify estimates were only 0.16 and 0.10, respectively. These correlations were lower than our observed correlation between TPF and purity ($r \sim 0.4$).

Carter, Scott L., et al. "Absolute quantification of somatic DNA alterations in human cancer." *Nature biotechnology* 30.5 (2012): 413-421.

Zhang, Naiqian, et al. "Predicting tumor purity from methylation microarray data." *Bioinformatics* 31.21 (2015): 3401-3405.

We have edited the text (p.4): "For comparison, we also calculated the correlation of TCGA pathologist-reported purity with the genomics-inferred purity measures ABSOLUTE (Carter et al. 2012) and InfiniumPurify (Zhang et al. 2015) in BRCA. The correlations of TCGA-annotated purity vs. the ABSOLUTE and InfiniumPurify estimates were only 0.16 and 0.10, respectively. These correlations were lower than our observed correlation between TPF and purity ($r \sim 0.4$)."

4. Couldn't the results in Supplementary Figure S2, just be explained since in TCGA you have many more slides that are tumor vs. normal, which means the same for tiles, and as such, you will overpredict tumor?

During training, we use downsampling to balance the tumor and normal data during training. So bias in training data does not explain those results. This was described in the Methods section on the CNN architecture and training (p.12): "To mitigate the effects of label imbalance in tumor/normal classification, undersampling was performed during training by rejecting inputs from the larger class according to class imbalances, such that, on average, the CNN receives equal number of tumor and normal tiles as input."

5. Do the authors have any interpretation why the transfer learning results don't work for TP53 mutation status?

As described by Yu et al, who studied the associations of crafted cellular features to TP53 in lung adenocarcinoma, TP53 mutation status was associated with the pixel intensity distribution in the cytoplasm and specific texture features within tumor nuclei (Yu et al. 2017). We conjecture that it is more difficult to identify such textural patterns from the ImageNet-based transfer learning network because such textures differ so much from those within macroscopic ImageNet images. On the other hand, tumor/normal classification may be more related to cell shape and size, which are simpler variables more likely to have analogs within ImageNet images.

Kun-Hsing Yu, Gerald J. Berry, Daniel L. Rubin, Christopher Ré, Russ B. Altman, Michael Snyder,"Association of Omics Features with Histopathology Patterns in Lung Adenocarcinoma", Cell Systems, Volume 5, Issue 6, 2017, Pages 620-627.e3, ISSN 2405-4712.

Although we admit these ideas remain speculative, we have added the following text to the Discussion (p.10): "For example, (Yu et al 2017) reported that TP53 mutation status is associated with the pixel intensity distribution in the cytoplasm and specific texture features within tumor nuclei, and it is possible that such textures are not in ImageNet while tumor/normal classification may be more related to cell shape and size, which are simpler variables more likely to have analogs within ImageNet."

6. It is not clear if you can use from the same patients tiles for training and testing? This would get dangerously close to overfitting, especially as the authors first use their cancer/normal classifier to discard "normal tiles", that means that already a selection is made for tiles that are similar. Can the authors show that results would be similar if using tiles for testing from left out patients?

This concern was also brought up by reviewer 4: minor comment m1. Please read there for the full response. In brief, the original manuscript did split the data at the slide level rather than at the patient level. We have re-trained and tested the classifiers on patient level splits of data. The AUCs are compared in Figure S1, showing that splitting the data at the patient level made little difference to classification AUCs. We note also that all cross-classification analyses inherently

use different patients for training and testing, so none of those results could arise from use of slides from the same patients.

7. For the TP53 analysis, did the investigators distinguish between gain and loss of function TP53 mutations? From a functional perspective the models should be able to distinguish the difference between these two types of mutations, and without considering this, there could be a bias in the analysis.

As reported by Donehower et al (2019), in the TCGA database loss-of-function is by far the most prevalent type of TP53-mutation. Indeed, among tumor types with TP53 mutations (including the cohorts used in our study), 91% of tumors with a TP53 mutation not only demonstrate a loss of function mutation, but also have loss of function in both TP53 alleles. Thus gain-of-function cases contribute negligibly to the available data and results.

Donehower et al. , Integrated Analysis of TP53 Gene and Pathway Alterations in The Cancer Genome Atlas, Cell Reports, Volume 28, Issue 5, 2019, Pages 1370-1384.e5, ISSN 2211-1247

8. Most of the analyses report AUC and accuracy as performance metrics, however, in the last analysis on the separate breast cancer & colorectal cancer cohort, the authors report RMSE. This makes the results difficult to compare? What are the accuracy results of these classifiers, when using similar model for tumor purity?

AUC and accuracy are metrics to evaluate classification performance, and RMSE is a metric to assess regression performance. These metrics are necessarily specific to each task, as the last analysis section uses quantitative rather than categorical training data.

9. Similarly, what happens when you use models trained on TCGA data and test on the new cohorts with local ground truth?

We agree with the reviewer this is an important question. We have used the TCGA-trained BRCA and COAD tumor/normal classifiers on the 8 colon cancer ROIs. The TCGA slide-trained classifiers were inferior to the BreCaHAD tile-trained CNNs. We have added the following to the manuscript (p.8):

“As a comparison, we also calculated RMSEs between purity and TPF as predicted by the TCGA-trained BRCA and COAD classifiers on the colorectal set. These RMSEs were 45% and 39%, respectively. These values were inferior to the BreCaHAD-trained CNNs, indicating that nuclear annotations provide additional predictive value beyond the overall slide label.”

10. It seems the authors are using different CNN models for each analysis? Does this suggest that a single modeling framework or architecture using CNNs is not able to learn the different tasks?

As we have used one only a small number of architectures (mainly Inception) and training methods (transfer learning vs full training) in this study, it would be premature for us to make a definitive conclusion about their limits, especially when small perturbation to architectures can impact results. What we can concretely say is that the transfer-learning based architecture of Figure 1a achieves high AUCs for tumor/normal and subtype classification, but performs poorly for TP53 mutation prediction. TP53 prediction is improved if one replaces transfer learning with full training, even if one maintains the Inception v3 architecture. Therefore, these results only suggest the applicability of transfer-learning is task-specific. We considered other architectures for the BreCAHAD/colorectal analyses but it is difficult to compare these to the other problems because the quantity and quality of training data were very different. We have made two small edits regarding the effect of changes in architecture to the Results (p.6) and the Discussion (p.11):

P.6: “We compared the effect of minor modification to the architecture on tumor/normal self- and cross-classifications. If we just used the Inception v3 architecture without the additional dense layers (see Methods), the results were inferior (Figure S6). Our architecture (Figure 1a) achieved a slightly higher AUC on average (0.04 ± 0.068) compared to the original Inception V3 network (Wilcoxon signed rank test p-value $< 1e-20$).”

P.10: “As the feature representation of the CNNs using transfer learning is optimized for the image-net dataset, additional dense layers are necessary when analyzing H&E slides. Although we found that the architecture in Figure 1a achieves slightly higher AUCs than the original inception architecture without dense layers, the optimal architecture of the dense layers is an open research question.”

11. The results described on lines 198 till 215 are hard to follow including the associated figure, can this analysis be explained in different way/improved/etc.?

We have made a number of writing edits to clarify this section on p.8. See main text for details.

Specific comments:

1. Methods Line 413: Can the authors be more specific what they mean with “clinical subtype”. This is very vague? Do they refer to histopathological subtype?

Yes, the reviewer is correct here. We rephrased the Method section to be more clear (p.11):

“For all other tissues, TCGA provided single cohorts that spanned multiple histopathological subtypes designated by pathologist annotations. This information is available at TCGA as ‘clinical’ supplementary files (filenames formatted as ‘nationwidechildrens.org_clinical_patient_{CANCERTYPE}.txt’). Only histopathological subtype annotations with at least 15 samples were considered. Samples with ambiguous or uninformative annotations were not included.”

2. The authors use the Tensorflow specific format TFRecord, this should be properly defined and introduced for the reader that is not familiar. Especially in Figure 1, it would be more appropriate to use a more general term referring to this step.

The following paragraph was added to the “Materials and Methods” section, under step 7 of “pre-processing and transfer learning steps” (p.12):

“The TFRecord file format is a simple format for storing a sequence of binary records. TFRecord is TensorFlow’s native storage format and enables high data throughput which results in a more efficient model training pipeline.”

3. The BreCaHAD data set should be better introduced in the methods section.

We thank the reviewer for the comment. The BreCaHAD dataset is now described in the methods section in more detail (p.14).

Reviewers' Comments:

Reviewer #1:

Remarks to the Author:

The Authors have adequately addressed my comments. I have no further issues.

Reviewer #2:

Remarks to the Author:

This revised manuscript uses the image database from TCGA to assess AI approaches to achieve aims of interest in molecular pathology: 1) distinguishing normal from cancer; 2) assessing tumor purity and 3) detection of TP53 mutations. The overarching goal and novelty is to evaluate how universal features gleaned from one cancer type apply to others. The work is performed meticulously, yielding a descriptive output for each of many comparisons (perhaps an overwhelming number of comparisons, with all the attendant hazards of finding results by chance). The paper will be of some interest to many researchers and of great interest to those with AI focus. As this is a "landscape" paper, it enjoys the power of broad relevance and suffers from weaknesses in detail, which are impossible to provide, given the scope of the work. The work could prompt infinite requests for revisions and additional analyses; therefore, this review is purposefully limited.

Key questions / topics: the manuscript should provide more pointed insights in the Discussion related to: 1) value or lack thereof of applying AI methods across tumor types to gain insights and 2) the importance of studying cancers at different scales and with broad sampling. How can this work guide the field of Pathology AI? Could this work have future implications for use of TCGA as well as raising some concerns about past publications that used TCGA data? This might help move the field forward.

Agnostic classifiers provide few insights about biology, and references to such require either strong explanations or might be considered difficult to support.

Specific Points

"In particular, classifiers trained on most cancer types successfully predicted tumor/normal status in BLCA (AUC=0.98±0.02), UCEC (AUC=0.97±0.03), and BRCA (AUC=0.97±0.04), suggesting that these cancers most clearly display features universal across types." Did the authors apply these classifiers to the CPTAC samples of lung cancer to show discrimination of normal and cancer?

"These positive correlations are not simply due to distinguishing tiles in tumor slides from tiles in normal slides. Figures 5c and 5d are analogous to Figures 5a and 5b, but computed only over tumor slides. The results are nearly unchanged, indicating that they reflect biology within tumor images." Please consider why the comment that the comparability across scale implies something biological? If it does imply biology, what exactly is implied? Certainly, it speaks to something scalable about morphology, which in turn might argue for something about biology. However, this statement suggests that the paper has implications for understanding biology, which is still quite remote. More directly, the work relates to the performance of agnostic classifiers across magnitudes of scale. Yet, we know that tumors are immensely heterogenous.

Why not test cancer types with low frequencies of TP53 mutations, perhaps as a group to give a sufficient sample size, to see if there is sensitivity in detection of mutations? The authors state, "For example, our comparison of tile and nucleus-level approaches indicated that intercellular regions are useful in predicting tumor purity, but it is uncertain what specific features mediate this relationship. It is worth noting that such analyses would not be possible without mixtures of tumor and normal regions together within images." By analogy, there may be something missed by not including cancers with a broad range of frequency of mutations.

Mark Sherman

Reviewer #3:

Remarks to the Author:

no further comments

Reviewer #4:

Remarks to the Author:

The authors have addressed most of the concerns of this reviewer. Several points causing confusion have now been clarified and useful explanations have been added.

Unfortunately, this work is also now superseded by the recent publication of pan-cancer computational pathology paper in Nature Cancer (July 2020) by Fu et al. [<https://doi.org/10.1038/s43018-020-0085-8>] whereby the authors reported results for classification of tumor/normal and tumor-types as well as the prediction of a variety of genomic variables from histology images via deep learning.

Reviewer #5:

Remarks to the Author:

The authors have responded to all my comments.

Reviewer comments

We thank the editor and reviewers for these constructive comments. Individual responses are provided below in red. The manuscript has also been revised accordingly, with relevant new changes in red. Some additional changes have also been made to the manuscript to meet journal editorial requirements, and these are in pink.

Best regards,

Jeff Chuang

Reviewer #1 (Remarks to the Author):

The Authors have adequately addressed my comments. I have no further issues.

We thank the reviewer for the feedback.

Reviewer #2 (Remarks to the Author):

This revised manuscript uses the image database from TCGA to assess AI approaches to achieve aims of interest in molecular pathology: 1) distinguishing normal from cancer; 2) assessing tumor purity and 3) detection of TP53 mutations. The overarching goal and novelty is to evaluate how universal features gleaned from one cancer type apply to others. The work is performed meticulously, yielding a descriptive output for each of many comparisons (perhaps an overwhelming number of comparisons, with all the attendant hazards of finding results by chance). The paper will be of some interest to many researchers and of great interest to those with AI focus. As this is a “landscape” paper, it enjoys the power of broad relevance and suffers from weaknesses in detail, which are impossible to provide, given the scope of the work. The work could prompt infinite requests for revisions and additional analyses; therefore, this review is purposefully limited.

We thank the reviewer for this thoughtful feedback.

Key questions / topics: the manuscript should provide more pointed insights in the Discussion related to: 1) value or lack thereof of applying AI methods across tumor types to gain insights and 2) the importance of studying cancers at different scales and with broad sampling. How can this work guide the field of Pathology AI? Could this work have future implications for use of TCGA as well as raising some concerns about past publications that used TCGA data? This might help move the field forward. Agnostic classifiers provide few insights about biology, and references to such require either strong explanations or might be considered difficult to support.

As the reviewer's questions pertain to value judgments about major issues in the field, we respond here somewhat subjectively. Regarding point (1), it is true that AI methods for known tasks have various non-optimality related to the datasets, which we have described

quantitatively in the Results section. However, more importantly, recent data and software engineering advances have primed the field for rapid conceptual and algorithmic growth. Therefore we believe that AI analyses combining tumor datasets can be valuable in many ways, but that open exploration is needed to realize this value. For comparison, when TCGA first began releasing sequencing data, they were soon criticized as having found little that wasn't already known, e.g. a prevalence of TP53 mutations. But over the next decade computational analysis concepts advanced greatly. Important non-obvious discoveries have come from multi-tumor-type analyses of TCGA sequence data, e.g. the relationship between MHC Class I-neoantigens and cytolytic activity (Rooney et al. 2015 Cell 160:48) and the identification of associations between leukocyte subsets and cancer survival (Gentles et al. 2015 Nature Medicine 21:938), among many others. We believe that multi-tumor-type AI analyses of even current image sets have similar untapped value. For this reason, we have chosen to focus the Discussion on conceptual directions that may be fruitful for data analysis, rather than uncertain estimates of data limitations. We have added a sentence summarizing this idea to the Discussion section ("Thus constraints of the TCGA data, such as a lack of FFPE non-tumor images and variable sample quality, do not preclude the development of effective classifiers, though new algorithmic concepts will be essential for individual image analysis questions just as they have been for realizing the value of TCGA sequence data, e.g. Rooney et al 2015, Gentles et al 2015). Regarding point (2), we agree that cancers should be imaged, and most importantly systematically assessed and annotated by pathologists, at different scales. We previously mentioned this in the Discussion ("Finally, to improve tile-level understanding using these approaches, further fine-grained pathological annotations with concomitant hypothesis development from the community are vital, e.g. through extended curation of TCGA and other sets. "). We have not added specific suggestions because we believe that annotation concepts are best designed by pathologists, while our team's primary expertise is from the data mining perspective.

Specific Points

"In particular, classifiers trained on most cancer types successfully predicted tumor/normal status in BLCA (AUC=0.98±0.02), UCEC (AUC=0.97±0.03), and BRCA (AUC=0.97±0.04), suggesting that these cancers most clearly display features universal across types." Did the authors apply these classifiers to the CPTAC samples of lung cancer to show discrimination of normal and cancer?

Yes, we previously applied the TCGA-trained BRCA, BLCA, and UCEC classifiers to the CPTAC data for LUAD and LUSC cancers. The results are in Figure 6 (this was in the prior revision, so the figure has not been modified). There was consistency between the test (TCGA) and validation (CPTAC) AUCs for these and other classifiers, as indicated by the strong correlation of values. We also note that the reviewer's question swaps the role of test and training sets. Our prior results showed the strong classifiability of BLCA, UCEC, and BRCA as test sets, but we did not discuss their value as training sets. Intuitively, a good test set should also be a good training set, but this relationship is affected by the balance of data quality and quantity. To more fully address the reviewer's concern, we trained classifiers using LUAD and LUSC cancers of CPTAC and applied them to TCGA data. There was a high correlation

between TCGA-trained and CPTAC-trained cross-classification AUCs, illustrating that the conclusions based on TCGA-trained classifiers are robust (Figure S8). In particular, BLCA, BRCA, and UCEC test sets exhibited high cross-classification AUCs when the CPTAC-trained classifiers were applied. We have added the following text and a novel supplemental figure S8 to the section on Validation of cross-classification relationships using CPTAC images:

“The CPTAC LUAD and LUSC datasets were also used to train classifiers, which were then tested on the TCGA cancer sets. We observed high correlation between TCGA-trained and CPTAC-trained cross-classification AUCs (Figure S8, LUAD: $r=0.98$, LUSC: $r=0.90$).”

“These positive correlations are not simply due to distinguishing tiles in tumor slides from tiles in normal slides. Figures 5c and 5d are analogous to Figures 5a and 5b, but computed only over tumor slides. The results are nearly unchanged, indicating that they reflect biology within tumor images.” Please consider why the comment that the comparability across scale implies something biological? If it does imply biology, what exactly is implied? Certainly, it speaks to something scalable about morphology, which in turn might argue for something about biology. However, this statement suggests that the paper has implications for understanding biology, which is still quite remote. More directly, the work relates to the performance of agnostic classifiers across magnitudes of scale. Yet, we know that tumors are immensely heterogenous.

The sentence in question explained that the results reflected not just systematic differences between tumor slides and non-tumor slides, but also intra-slide effects. As this finding is data-driven but not mechanistic, we have adjusted the wording “biology within tumor images” to “behavior within tumor images.”

Why not test cancer types with low frequencies of TP53 mutations, perhaps as a group to give a sufficient sample size, to see if there is sensitivity in detection of mutations? The authors state, “For example, our comparison of tile and nucleus-level approaches indicated that intercellular regions are useful in predicting tumor purity, but it is uncertain what specific features mediate this relationship. It is worth noting that such analyses would not be possible without mixtures of tumor and normal regions together within images.” By analogy, there may be something missed by not including cancers with a broad range of frequency of mutations.

We agree this is an interesting question, but a challenge is that other cancer types often have very low fractions of TP53 mutations meeting the quality control criteria. For instance, the total fractions of mutant TP53 samples for TCGA-PRAD and TCGA-KIRC are 0.24 and 0.15 respectively. However, these fractions drop to 0.07 and 0.02 under our requirement that the TP53 mutations have impact MODERATE or HIGH, fractions which are far too low to train effective deep learning models. To obtain sufficient numbers of TP53 mutant samples, we would have to aggregate many cancer types together. This would then necessitate combinatorial analysis of cross-classifiability relationships, a challenge with a number of uncertainties. Such work is a significant effort that we believe is of more appropriate size for a separate manuscript.

Reviewer #3 (Remarks to the Author):

no further comments

We thank the reviewer for the feedback.

Reviewer #4 (Remarks to the Author):

The authors have addressed most of the concerns of this reviewer. Several points causing confusion have now been clarified and useful explanations have been added.

Unfortunately, this work is also now superseded by the recent publication of pan-cancer computational pathology paper in Nature Cancer (July 2020) by Fu et al. [<https://doi.org/10.1038/s43018-020-0085-8>] whereby the authors reported results for classification of tumor/normal and tumor-types as well as the prediction of a variety of genomic variables from histology images via deep learning.

The key difference with the Fu et al. work is that we focus on cross-classifiability across tumor types. Fu et al did not investigate this issue. Our discussion section noted this difference with respect to the biorxiv preprint by Fu et al (their Nature Cancer publication came out while our paper was in review). To make this clearer, we have updated the citation to reference the Fu et al Nature Cancer publication. Incidentally, with a few exceptions the genomic variables predicted by Fu et al have moderate AUC values. This supports the need for further investigation of approaches-- such as cross-classification-- that could improve predictions.

Reviewer #5 (Remarks to the Author):

The authors have responded to all my comments.

We thank the reviewer for the feedback.